# Age-related changes in P2Y receptor signalling in mouse cochlear supporting cells

Sarah A. Hool[1] 🆔, Jing-Yi Jeng[1] 🆔, Daniel J. Jagger[2] 🆔, Walter Marcotti[1,3] 🆔 and Federico Ceriani[1] 🆔

[1]*School of Biosciences, University of Sheffield, Sheffield, UK*
[2]*UCL Ear Institute, University College London, London, UK*
[3]*Neuroscience Institute, University of Sheffield, Sheffield, UK*

Handling Editors: Katalin Toth & Samuel Young

The peer review history is available in the Supporting Information section of this article (https://doi.org/10.1113/JP284980#support-information-section).

**Abstract** Our sense of hearing depends on the function of a specialised class of sensory cells, the hair cells, which are found in the organ of Corti of the mammalian cochlea. The unique physiological environment in which these cells operate is maintained by a syncitium of non-sensory supporting cells, which are crucial for regulating cochlear physiology and metabolic homeostasis. Despite their importance for cochlear function, the role of these supporting cells in age-related hearing loss, the most common sensory deficit in the elderly, is poorly understood. Here, we investigated the age-related changes in the expression and function of metabotropic purinergic

**Sarah Hool** received her BSc in Biomedical Science, and MSc in Clinical Neurology, at the University of Sheffield (UK). She is now a PhD student in the Hearing Research Group (https://www.sheffield.ac.uk/hearing) at the University of Sheffield. Her research investigates age-related changes in the physiological and morphological properties of supporting cells in the mammalian cochlea, with the overall aim of furthering our understanding of the mechanisms involved in age-related hearing loss.

The Journal of Physiology

receptors ($P2Y_1$, $P2Y_2$ and $P2Y_4$) in the supporting cells of the cochlear apical coil. Purinergic signalling in supporting cells is crucial during the development of the organ of Corti and purinergic receptors are known to undergo changes in expression during ageing in several tissues. Immuno-labelling and $Ca^{2+}$ imaging experiments revealed a downregulation of P2Y receptor expression and a decrease of purinergic-mediated calcium responses after early postnatal stages in the supporting cells. An upregulation of P2Y receptor expression was observed in the aged cochlea when compared to 1 month-old adults. The aged mice also had significantly larger calcium responses and displayed calcium oscillations during prolonged agonist applications. We conclude that supporting cells in the aged cochlea upregulate $P2Y_2$ and $P2Y_4$ receptors and display purinergic-induced $Ca^{2+}$ responses that mimic those observed during pre-hearing stages of development, possibly aimed at limiting or preventing further damage to the sensory epithelium.

(Received 4 May 2023; accepted after revision 16 August 2023; first published online 4 September 2023)

**Corresponding author** F. Ceriani: School of Biosciences, University of Sheffield, Sheffield, S10 2TN, UK. Email: f.ceriani@sheffield.ac.uk

**Abstract figure legend** We investigated the progressive changes in metabotropic purinergic signalling in the supporting cells of murine cochlear sensory epithelia. Nanomolar levels of ATP induce $Ca^{2+}$ responses in the supporting cells of pre-hearing mice, which are mediated by $P2Y_1$, $P2Y_2$ and $P2Y_4$ receptors. Adult supporting cells downregulate P2Y receptors and lack $Ca^{2+}$ responses. Aged supporting cells upregulate ATP-induced $Ca^{2+}$ responses mediated by P2Y2 and P2Y4 receptors. Calcium responses in the aged cochlea resemble those in the developing cochlea, possibly limiting or preventing further damage to the sensory epithelium.

## Key points

- Age-related hearing loss is associated with lower hearing sensitivity and decreased ability to understand speech.
- We investigated age-related changes in the expression and function of metabotropic purinergic (P2Y) receptors in cochlear non-sensory supporting cells of mice displaying early-onset (C57BL/6N) and late-onset (C3H/HeJ) hearing loss.
- The expression of $P2Y_1$, $P2Y_2$ and $P2Y_4$ receptors in the supporting cells decreased during cochlear maturation, but that of $P2Y_2$ and $P2Y_4$ was upregulated in the aged cochlea.
- $P2Y_2$ and $P2Y_4$ receptors were primarily responsible for the ATP-induced $Ca^{2+}$ responses in the supporting cells.
- The degree of purinergic expression upregulation in aged supporting cells mirrored hearing loss progression in the different mouse strains.
- We propose that the upregulation of purinergic-mediated signalling in the aged cochlea is subsequent to age-related changes in the hair cells and may act as a protective mechanism to limit or to avoid further damage to the sensory epithelium.

## Introduction

Age-related hearing loss (ARHL), also known as presbycusis, is one of the most common chronic conditions in the elderly. It is associated with progressive changes in the peripheral and central auditory system leading to hearing impairment, difficulty in understanding speech and impaired sound localisation (Bowl & Dawson, 2019; Gates & Mills, 2005). The severity of ARHL is compounded by its association with several comorbidities, including depression and declining cognitive abilities (Livingston et al., 2020), greatly worsening the social impact of the disease. Although changes to central auditory processing may contribute to the pathology (Bao et al., 2020; Frisina 2009), the most prevalent causes of ARHL have been linked to the cochlea (Schuknecht & Gacek, 1993). The sensory epithelium of the mammalian cochlea, the organ of Corti, contains the sensory inner hair cells (IHCs) and outer hair cells (OHCs), along with several highly differentiated and functionally distinct non-sensory supporting cells (Gale & Jagger 2010; Wan et al., 2013). Although most of the research into the molecular, morphological and functional changes occurring in the ageing cochlea has focused on the hair

cells and their innervation, little is known about the role of supporting cells in the pathophysiology of cochlear ageing (Jeng et al., 2021; Jeng, Ceriani et al., 2020; Jeng, Johnson et al., 2020; Lauer et al., 2012; Liu et al., 2022; Sergeyenko et al., 2013; Viana et al., 2015; Zachary & Fuchs, 2015).

The supporting cells located in the medial compartment are organised in a single layer, which develops from a pre-hearing structure called the greater epithelial ridge (GER) (Fritzsch & Elliott, 2018; Gale & Jagger 2010). Supporting cells are key for normal hearing since they are responsible for maintaining the environment of the cochlear partition. This role includes the establishment and maintenance of the ionic separation between the different cochlear compartments that underpins the endocochlear potential (Gulley & Reese 1976; Wangemann 2006), the recycling of potassium ions through gap-junction networks (Jagger & Forge, 2006; Kikuchi 2000; Spicer & Schulte 1998), and the removal of glutamate around the IHCs (Glowatzki et al., 2006). During pre-hearing stages, the supporting cells have been shown to regulate the development and fine tuning of sensory hair cells and their innervation via purinergic signalling (e.g. Babola et al., 2018; Ceriani et al., 2019; Johnson et al., 2017; Tritsch et al., 2007). ATP release from supporting cells activates G-coupled P2Y autoreceptors located on their endolymphatic surface, promoting $Ca^{2+}$ release from intracellular stores (Rabbit & Holman 2021) and inducing $Ca^{2+}$ oscillations when applied at sub-micromolar concentrations (Ceriani et al., 2016; Gale et al., 2004; Piazza et al., 2007). Calcium elevations propagate in a wave-like fashion in the gap-junction-coupled syncytium of supporting cells (Anselmi et al., 2008; Beltramello et al., 2005; Ceriani et al., 2016; Majumder 2010), similar to the $Ca^{2+}$ waves recorded in glial cells (Newman 2001). Although this mechanism is well established, the identity of the P2Y receptor subtypes mediating these $Ca^{2+}$ waves is still unclear, with evidence supporting either $P2Y_2$ and $P2Y_4$ (Huang et al., 2010; Piazza et al., 2007) or $P2Y_1$ (Babola et al., 2020). Although purinergic-induced $Ca^{2+}$ signalling has also been observed in mature cochlear explants (Chan & Rouse 2016; Horvath et al., 2016; Sirko et al., 2019), P2Y expression is downregulated when the epithelium reaches functional maturity after the onset of hearing (Huang et al., 2010).

Considering that P2Y receptors are associated with age-related functional changes in other tissues (Erb et al., 2015; Gao et al., 2019; Iring et al., 2022; Reichenbach & Bringmann, 2016; Wallace et al., 2006), we hypothesised that they could also play a role in the altered functions of the aged cochlea. In this study, we used different mouse lines displaying varying degrees of progressive hearing loss to identify possible changes in the distribution and functional properties of P2Y receptors in the supporting cells of the ageing mouse.

Immunolabelling experiments showed that supporting cells of the cochlear inner sulcus upregulate P2Y receptors in aged mice. Using ratiometric $Ca^{2+}$ imaging, we showed that the application of nanomolar concentrations of ATP elicited $Ca^{2+}$ responses in the supporting cells. We also demonstrated that $Ca^{2+}$ responses mediated by $P2Y_2$ and $P2Y_4$, but not $P2Y_1$ receptors increase with age especially in mice affected by early-onset hearing loss.

## Methods

### Ethics statement

All procedures were approved by the Home Office, in line with the Animals (Scientific Procedures) Act 1986, and were approved by the Ethical Review Committee at the University of Sheffield (180 626_Mar). Mice had an unlimited access to food and water. Schedule 1 cervical dislocation, and subsequent decapitation, was used to humanely kill mice used for *ex vivo* experiments. For *in vivo* recordings of auditory brainstem responses (ABRs), mice were anaesthetised using an intraperitoneal (I.P.) injection of ketamine (100 mg kg$^{-1}$ body weight; Fort Dodge Animal Health, Fort Dodge, IA, USA) and xylazine (10 mg kg$^{-1}$ body weight, Rompun 2%; Bayer HealthCare LLC, Tarrytown, NY, USA). At the end of the ABR experiments, mice were either killed by cervical dislocation or recovered from anaesthesia (I.P. injection of atipamezole: 1 mg kg$^{-1}$ body weight). During the recovery from anaesthesia, mice were returned to their cage placed on a thermal mat and monitored over the following 2–5 h. Littermate mice of either sex were used for experiments.

### Mouse strains

We used three strains of mice known to display differing severities of progressive hearing loss. C57BL/6N (6N) exhibit early onset hearing loss due to a hypomorphic allele in *Cadherin 23* (*Cdh23$^{ahl}$*) (Jeng et al., 2021; Jeng, Ceriani et al., 2020; Jeng, Johnson et al., 2020; Johnson et al., 1997; Kane et al., 2012; Noben-Trauth et al., 2003). These mice show early onset hearing loss that is already evident at 3 months of age in the high-frequency region and continues to worsen over time (Jeng, Ceriani et al., 2020). *Cdh23* encodes for cadherin-23 that, together with protocadherin-15, forms the tip links that are required to gate the mechano-electrical transducer channels (Kazmierczak et al., 2007). The C57BL/6N$^{Cdh23+}$ (6N-Repaired) strain is co-isogenic to 6N, except for having their *Cdh23* mutation corrected via CRISPR/Cas9-mediated homology directed repair (Mianné et al., 2016). 6N-Repaired mice retain normal high-frequency hearing thresholds until late in life, but they still exhibit low-frequency hearing loss comparable

to 6N mice (Jeng, Ceriani et al., 2020). The third strain is C3H/HeJ (C3H), which is known to retain good hearing function until later in life (Jeng et al., 2021; Jeng, Ceriani et al., 2020; Jeng, Johnson et al., 2020; Ohlemiller et al., 2016).

### Auditory brainstem responses

Following the onset of anaesthesia (see Ethics statement above) and the loss of the retraction reflex with a toe pinch, mice were placed onto a heat mat (37°C) in a soundproof chamber (MAC-3 acoustic chamber, IAC Acoustic, UK). Subdermal electrodes were placed under the skin behind the pinna of each ear (reference and ground electrode) and on the vertex of the mouse (active electrode) as previously described (Ingham et al., 2011). Auditory brainstem responses (ABRs) were recorded from male and female mice from the strains listed above. Sound stimuli were delivered to the ear by calibrated loudspeakers (MF1-S, Multi Field Speaker, Tucker-Davis Technologies, USA) placed 10 cm from the animal's pinna. Sound pressure was calibrated with a low-noise microphone probe system (ER10B+, Etymotic, USA). Experiments were performed using a customised software (Ingham et al., 2011) driving an RZ6 auditory processor (Tucker-Davis Technologies). Auditory thresholds were estimated from the resulting ABR waveform and defined as the lowest sound level (measured in decibel, dB) where any recognisable feature of the waveform was visible. Responses were measured for clicks and stimulus pure tones of frequencies at 3, 6, 12, 18, 24, 30, 36 and 42 kHz. Stimulus sound pressure levels were typically 0–95 dB SPL, presented in steps of 5 dB SPL. The brainstem response signal was averaged over 256 repetitions. Tone bursts were 5 ms in duration with a 1 ms on/off ramp time, which was presented at a rate of 42.6/s.

### Immunofluorescence microscopy

Inner ears were dissected out and perfused with 4% paraformaldehyde solution for 20 min at room temperature. The fixed samples were then micro-dissected to retrieve the apical turn of the organ of Corti. These were then blocked in 5% horse serum for an hour at room temperature before being incubated at 35°C overnight in primary antibody solution. The primary antibodies used included anti-$P2Y_1$ (Alomone Labs, cat. no. APR-009), anti-$P2Y_2$ (Alomone Labs, cat. no. APR-010) and anti-$P2Y_4$ (Alomone Labs, cat. no. APR-006) all at a concentration of 1:800. The following day, samples were washed in phosphate buffer saline (PBS) and incubated for an hour at 35°C in secondary antibody solution. This solution contained goat anti-rabbit IgG 490 (Alexa fluor 488, A11034, Thermo Fisher) and Texas Red-X Phalloidin for staining of F-actin (1:1000, T7471, Thermo Fisher). The samples were then washed in PBS and mounted

in VECTASHIELD (H-1000). Images of the cochleae were taken with a Zeiss LSM 880 Airyscan microscope. The *z*-stack images were taken with 0.5 $\mu$m incremental steps. Fiji ImageJ software was used to process the images and generate maximum intensity projections (https://imagej.net/Fiji).

### Tissue preparation and dye loading

Experiments were performed from the 9–12 kHz region of the cochlear apical coil (Müller et al., 2005). The apical coil was dissected out in extracellular solution composed of (in mM): 135 NaCl, 5.8 KCl, 1.3 $CaCl_2$, 0.9 $MgCl_2$, 0.7 $NaH_2PO_4$, 5.6 D-glucose, 10 HEPES-NaOH. Sodium pyruvate (2 mM), amino acids and vitamins were added from concentrates (Thermo Fisher Scientific, UK). The pH was adjusted to 7.48 ($\sim$308 mmol kg$^{-1}$). Once dissected, the organ of Corti was transferred into a calcium dye loading solution and incubated at 37°C for 30 min. The loading solution contained DMEM/F12, 0.1% pluronic F-127 (Thermo Fisher Scientific, UK), 250 $\mu$m sulfinpyrazone to avoid dye sequestration (Sigma-Aldrich) and the cell-permeant fluorescent calcium indicator Fura-2-AM at a concentration of 10 $\mu$M (Thermo Fisher Scientific, UK).

### Calcium imaging

After loading the cochlea with the $Ca^{2+}$ dye, the dissected apical coil was transferred to a microscope chamber, immobilised via a nylon mesh attached to a stainless-steel ring, and then viewed with an upright microscope (Olympus BX51WI, Japan) equipped with Nomarski Differential Interference Contrast (DIC) optics (Olympus LUMPlanFl 60× water immersion objective, 1.0 NA) and a 15× eyepiece. The microscope chamber was continuously perfused with extracellular solution by a peristaltic pump (Cole-Palmer, UK). All experiments were performed at room temperature. Preparations were perfused with extracellular solution for 15 min before imaging to allow for dye de-esterification.

For pharmacological experiments, ATP, ADP, UTP, MRS2500 (Tocris Bioscience, UK), Thapsigargin (Tocris Bioscience, UK), AR-C 118925XX (Tocris Bioscience, UK), and MRS4062 (Tocris Bioscience, UK) were diluted from concentrates in extracellular solution and applied to the preparation using either a Pico-injector (PLI-100, Harvard Apparatus, USA) or a gravity-fed perfusion system through a multi-barrel microcapillary. Pressure was kept at a minimum to avoid motion during imaging and the triggering of mechanically induced $Ca^{2+}$ responses.

Fura-2 fluorescence was excited at two alternating wavelengths by using two light-emitting diodes (LEDs)

with peak wavelengths of 365 nm and 385 nm, respectively (Thorlabs, USA) filtered through narrow band filters (FF360/23 and FF392/23, respectively Semrock, USA). Fura-2 emission was separated from excitation light through a long-pass dichromatic mirror (T495lpxr, Chroma, USA) and filtered through a bandpass interference filter (ET525/50M, Chroma). Fluorescence images were formed on a scientific grade CMOS camera (Hamamatsu ORCA Flash 4.0 V2, Japan) operating in 'external trigger' mode and captured on a computer using Micro-Manager (Edelstein et al., 2014). The timing of camera exposure and LED illumination were controlled by a microcontroller (Arduino Mega, Italy) and set by custom-built Python software (Python 3.10, Python Software Foundation). LED light was switched on only during frame exposure by the camera to minimise phototoxicity. The exposure time was set at 65 ms for the 365 nm LED and 25 ms for the 385 nm LED. The exposures at the two wavelengths were separated by 100 ms and repeated every 1 s (final framerate for the ratio signal: 1 Hz). The framerate was chosen to minimise UV light exposure to the cells, while maintaining enough temporal resolution to sample the relatively slow purinergic-induced $Ca^{2+}$ changes in supporting cells. A brightfield image of the preparation was generally acquired at the end of each recording to aide with regions of interest (ROIs) placement (see below).

Images were analysed off-line using software and scripts written in Python and ImageJ (NIH). Square ROIs (20 × 20 pixels, or 3.5 $\mu$m × 3.5 $\mu$m) were manually placed using a custom software on identified supporting cells in the proximity of hair cells (Fig. 5) using an average of fluorescence images captured using 385 nm excitation or a brightfield image as reference. The background fluorescence value was calculated as the average pixel value from a timelapse acquisition in a cochlear explant not loaded with the Fura-2 $Ca^{2+}$ dye and subtracted from each fluorescence image.

$Ca^{2+}$ changes were quantified as fractional dye emission ratio changes:

$$\frac{\Delta R}{R_0} = \frac{R(t) - R_0}{R_0},$$

where $R(t)$ is the ratio between the fluorescence emission values for excitation at 365 nm and 385 nm at time $t$. $R_0$ denotes the baseline $R$ value calculated as an average during the 3 s before stimulation.

To quantify $Ca^{2+}$ responses, we measured the maximum and average of the $\Delta R/R_0$ signal from individual ROIs during the application of the different compounds. To compare recordings in which compounds were applied for different durations, we limited the analysis to the first 30 s after the onset of drug application. Calcium oscillations were identified by detecting local maxima in the $Ca^{2+}$ trace using a peak-finding algorithm (function *find_peaks* of the scipy.signal Python module). The frequency was calculated as the number of $Ca^{2+}$ oscillations divided by the duration of the stimulus. Individual traces were subsequently inspected manually to remove any spurious peaks and re-analysed.

## Statistical analysis

Statistical comparisons between two paired groups were made by Wilcoxon signed-rank test. For statistical comparisons of unpaired data, analysis of variance (one-way or two-way ANOVA followed by an appropriate *post hoc* test), the non-parametric Kruskal-Wallis test (followed by pairwise Wilcoxon rank-sum test with Bonferroni correction for multiple comparisons) or the non-parametric aligned ranks transformation (ART) two-way ANOVA (followed by Tukey's *post hoc* test or the pairwise Wilcoxon rank-sum test with Bonferroni correction) were used. $P < 0.05$ was selected as the criterion for statistical significance. Statistical tests were performed using R software (R Core Team 2022). Only mean values with a similar variance between groups were compared. Values are quoted in text and figures as means ± SD. Animals of either sex were randomly assigned to the different experimental groups. No statistical methods were used to define sample size, which was determined based on previous published similar work on cochlear ageing and supporting cell physiology (e.g. Ceriani et al., 2016; Ceriani et al., 2019; Jeng et al., 2021; Jeng, Ceriani et al., 2020; Jeng, Johnson et al., 2020). Animals were taken from several cages and breeding pairs over a period of several months.

## Results

### Progression of ABR thresholds between C57BL/6N and C3H/HeJ mice

Auditory brainstem responses (ABRs) were used to test hearing sensitivity of 1-month and 20-month-old C57Bl/6N (6N) and C3H/HeJ (C3H) mice (Fig. 1*A* and *B*). Thresholds for clicks recorded from 6N mice largely increased between young adult and aged mice ($P < 0.0001$, *post hoc* test from one-way ANOVA), but not in C3H mice ($P = 0.7607$) (Fig. 1*C*). Pure-tone evoked ABRs thresholds (Fig. 1*D* and *E*) were also found to increase with age in 6N mice ($P < 0.0001$: two-way ANOVA) but less so in C3H mice ($P = 0.0155$). These data corroborate previous finding indicating that the hypomorphic allele in $Cdh23^{ahl}$ present in 6N mice, leads to the almost complete loss of hearing function in aged mice (Jeng et al., 2021; Jeng, Ceriani et al., 2020; Jeng, Johnson et al., 2020; Johnson et al., 1997;

Kane et al., 2012; Noben-Trauth et al., 2003). However, a recent study has shown that the correction of the *Cdh23^{ahl}* allele using CRISPR/Cas9 (Mianné et al., 2016) was able to rescue high-frequency, but not low-frequency hair cell loss in the aged co-isogenic 6N-Repaired mice (<18 kHz: Jeng, Ceriani et al., 2020). This indicates that other changes, rather than the *Cdh23^{ahl}* allele, are likely to be responsible for the progressive low-frequency hearing loss in the 6N mouse strain. Since supporting cells are essential for the normal development, survival and function of hair cells and their neurons (e.g. Jagger & Forge, 2015; Wan et al., 2013), we investigated whether they undergo strain-specific changes in the ageing cochlea.

## Changes in purinergic receptor expression in the ageing cochlea

In the mature cochlea, supporting cells are divided into two compartments delimited by the tunnel of Corti: the medial compartment, where the IHCs are located and the outer compartment, which contains the OHCs (Jagger & Forge, 2006). The supporting cells located in the medial compartment are organised in a single layer and include the inner phalangeal cells (IPhC) and the inner border cells (IBC), which surround the IHCs, and the cells forming the so-called inner sulcus (IS, Fig. 2*A*) (Fritzsch & Elliott, 2018). This area develops from the greater epithelial ridge (GER), which contains the Kölliker's organ that is transiently present during pre-hearing developmental stages (Fig. 2*A*). Conversely, supporting cells in the lateral region of the organ of Corti are part of the lesser epithelial ridge (LER) or outer sulcus.

Among the different types of P2Y receptors expressed in the mammalian cochlea, P2Y$_1$, P2Y$_2$, and P2Y$_4$ have been shown to be the primary isoforms associated with the generation of both spontaneous and induced Ca$^{2+}$ signals (Babola et al., 2020; Huang et al., 2010; Piazza et al., 2007; Prades et al., 2021). Therefore, we investigated whether the level of expression and localization of these three P2Y receptors change in the apical-coil of the cochlear sensory epithelium (6−12 kHz, Müller et al., 2005) from ageing mice using immunostaining. P2Y$_1$ receptors were highly expressed in the supporting cells of the GER during pre-hearing stages of development (P7) in both 6N (Fig. 2*B*, *D* and *F*, top row) and C3H mice (Fig. 2*C*, *E* and *G*, top row). However, compared to P7, the number of P2Y$_1$ puncta were largely decreased in the supporting cells of young adult (1 month) and aged mice from both 6N (Fig. 2*D* and *F*) and C3H mice (Fig. 2*E* and *G*). P2Y$_2$ and P2Y$_4$ puncta were also largely reduced between P7 and 1 month of age in both 6N (P2Y$_2$: Fig. 3*A* and *C*; P2Y$_4$: Fig. 4*A* and *C*) and C3H mouse strains (P2Y$_2$: Fig. 3*B* and *D*; P2Y$_4$: Fig. 4*B* and *D*). However, P2Y$_2$ and P2Y$_4$ puncta reappeared in the supporting cells of 6N aged mice (P2Y$_2$: Fig. 3*C*; P2Y$_4$: Fig. 4*C*). These

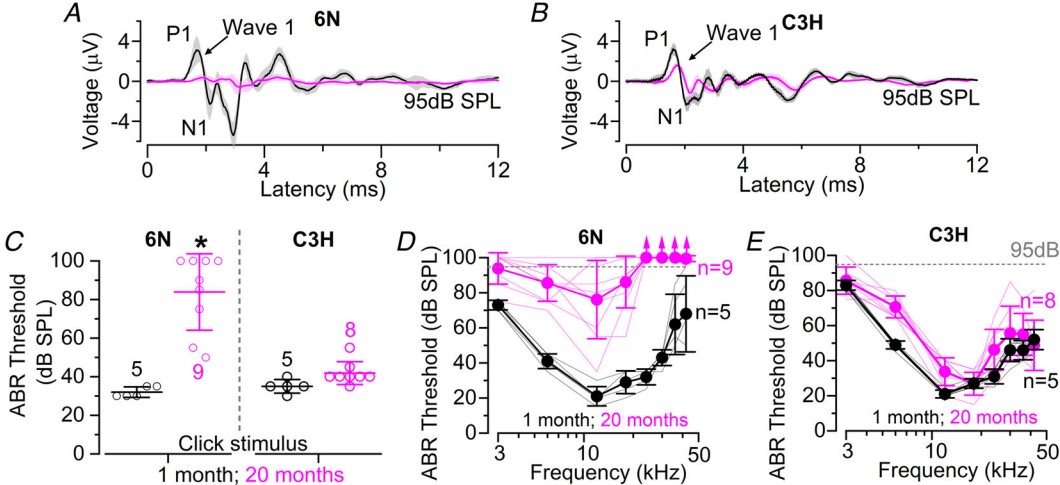

**Figure 1. ABR thresholds increase in ageing 6N but not C3H mice**
*A* and *B*, average auditory brainstem response (ABR) waveforms at 12 kHz and 95 dB SPL at 1 month and 20 months of age in 6N (*A*) and C3H (*B*) mice. Continuous lines represent the average values from all recordings and the shaded areas the SD. The number of mice is as listed in panel *C* below). P1 and N1 indicate the positive and negative peaks of wave 1, respectively, which represent the signal from the afferent fibres connecting with the IHCs. *C*, average ABR thresholds elicited by click stimuli applied to 6N (left) and C3H (right) mice at 1 month and 20 months of age. Number of mice used is shown above or below the averages and single data points (plotted as open circles). Significance values are indicated by the asterisks ($P <$ 0.0001, one-way ANOVA). *D* and *E*, ABR thresholds for frequency-specific pure tone stimulations (3, 6, 12, 18, 24, 36, 42 kHz) recorded from 6N (*D*) and C3H (*E*) mice at 1 month and 20 months of age. The numbers next to the traces represents the mice tested for each age/strain. Data are shown as mean ± SD (single animal recoding plotted as thinner lines). The dashed line represents the upper threshold limit used for ABRs (95 dB).

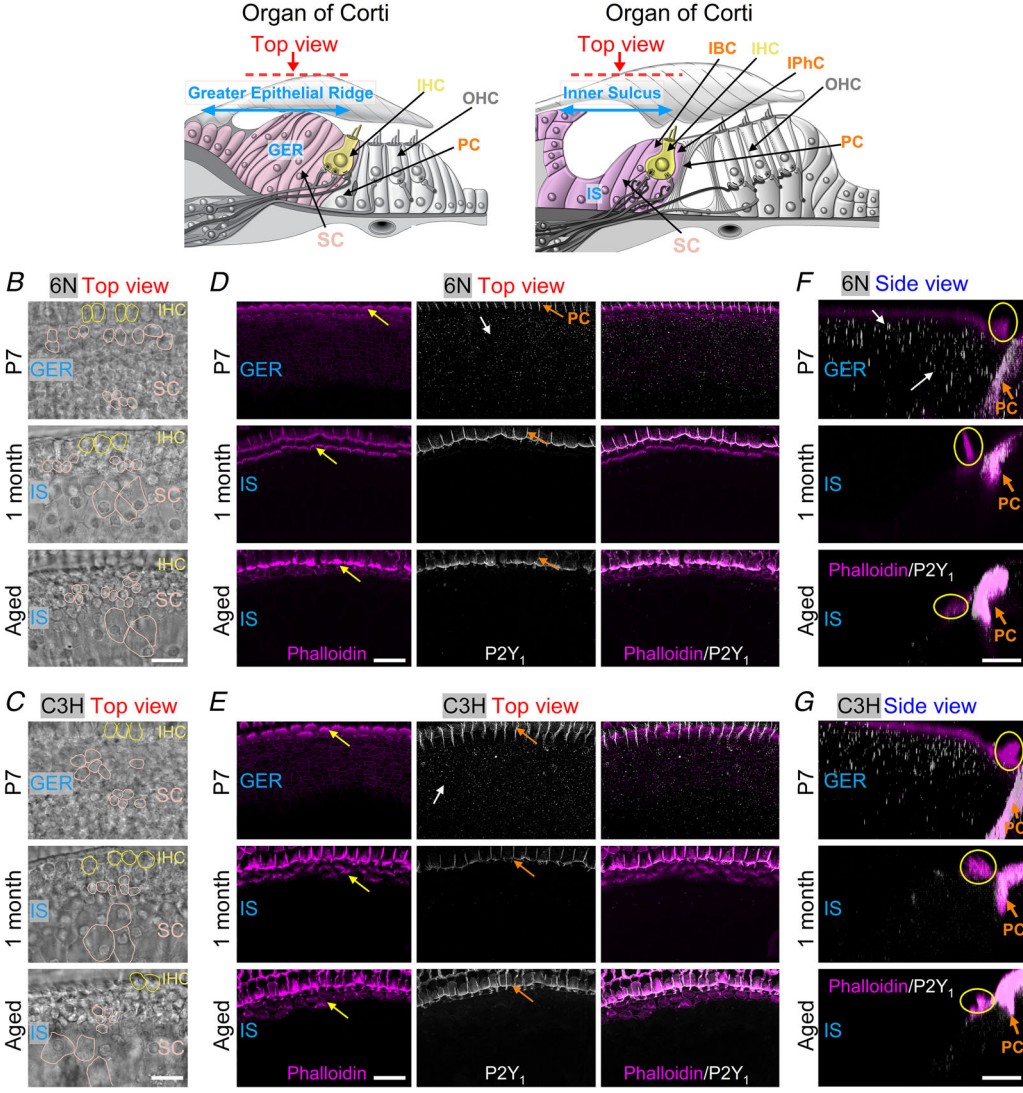

**Figure 2. Expression of P2Y₁ receptors in the ageing cochlea from 6N and C3H mice**

*A*, diagram showing a cross-section side-view of a pre-hearing (left) and mature (right) organ of Corti. IHC: inner hair cell; OHC: outer hair cell; IPhC: inner phalangeal cell; IBC: inner border cell; PC: pillar cell; GER: greater epithelial ridge; IS: inner sulcus; SC: supporting cells of the GER and IS, which also include IPhCs and IBCs. The 'Immature' and 'Mature' designations refer to before and after the onset or hearing, respectively, which in mice occurs at around P12. The images below represent either the top view (red arrow) or the side view of the GER and IS. The image on the left was modified from Ceriani et al. (2019). *B* and *C*, brightfield images of the organ of Corti in P7 (top), 1-month-old (middle) and aged (bottom) 6N (*B*) and C3H (*C*) mice taken from the top-view orientation used for the fluorescence images in panels *D* and *E*; these images highlight the location of the different supporting cell types (SC) in the GER and IS and the IHCs. Note that for simplicity, only a few SCs are highlighted in panels *B* and *C*. Scale bars are 20 $\mu$m. *D* and *E*, maximum intensity projections of confocal *z*-stacks showing images of the GER and IS viewed from the top (red arrow in panel *A*) in P7 (top panels), 1-month-old (middle panels) and aged (17–21 months: bottom panels) 6N (*D*) and C3H (*E*) mice. Left columns show the actin-marker phalloidin (magenta), which is labelling the hair bundles of the IHCs (yellow arrows) and the membrane of some of the supporting cells. Middle columns show the P2Y₁ puncta-like labelling (white) in the supporting cells from the bulk of the GER and IS (white arrows). P2Y₁ was also expressed in the pillar cells (PCs: orange arrows) which were not investigated in this study. Right panels show the merged images. Scale bars are 20 $\mu$m. *F* and *G*, maximum intensity projections of confocal z-stack images of the GER and IS viewed from the side (see panel *A*). Phalloidin: magenta; P2Y₁: white. Note that phalloidin primarily labels the hair bundle of the IHCs (yellow circles) and the pillar cells (PC). Scale bars are 10 $\mu$m.

immunolabelling experiments provide a qualitative indication that the expression of the three P2Y receptors is likely to change during the maturation and ageing of the cochlea, and also between the 6N and C3H strains for $P2Y_2$ and $P2Y_4$.

### Age-related changes in purinergic-mediated $Ca^{2+}$ signalling in cochlear supporting cells

To obtain a functional readout of purinergic receptors in cochlear supporting cells, we performed ratiometric $Ca^{2+}$ imaging with the dye Fura-2 in acute explants of the apical-coil cochlear region (6–12 kHz) of 6N, 6N-Repaired and C3H mice. In the pre-hearing developing cochlea, supporting cells of the GER release ATP that, by binding to G-protein coupled P2Y auto-receptors, leads to the generation and propagation of intercellular $Ca^{2+}$ waves (Piazza et al., 2007; Tritsch et al., 2007). Although some previous studies have used micromolar concentrations of ATP to investigate purinergic

receptor activity in the GER (1–100 $\mu$m: Horváth et al., 2016; Lahne & Gale, 2008; Rabbitt & Holman, 2021; Tritsch & Bergles, 2010), we found that 100 nм ATP was sufficient to trigger $Ca^{2+}$ elevations in the supporting cells of the GER in the close proximity of the IHCs (Fig. 5).

Calcium dynamics in the GER and IS were measured from the supporting cells next to the IHCs (Fig. 6A). In 6N mice, we observed age-dependent changes in $Ca^{2+}$ dynamics following application of 100 nм ATP (Fig. 6B–E, N). Both sustained $Ca^{2+}$ oscillations and repetitive $Ca^{2+}$ 'spikes' (i.e. oscillations where $Ca^{2+}$ levels returned to baseline after every cyclic increase) following application of ATP were recorded from the supporting cells of the GER of P7 pre-hearing mice (Fig. 6B). However, both average and maximum $Ca^{2+}$ responses were greatly reduced in young adult mice (1 month-old, $P < 0.0001$ for both comparisons, pairwise Wilcoxon rank sum test, ART two-way ANOVA, Fig. 6O and P). This change in $Ca^{2+}$ response with cochlear maturation is consistent with previous studies showing a reduction of purinergic-induced responses (Tritsch & Bergles, 2010)

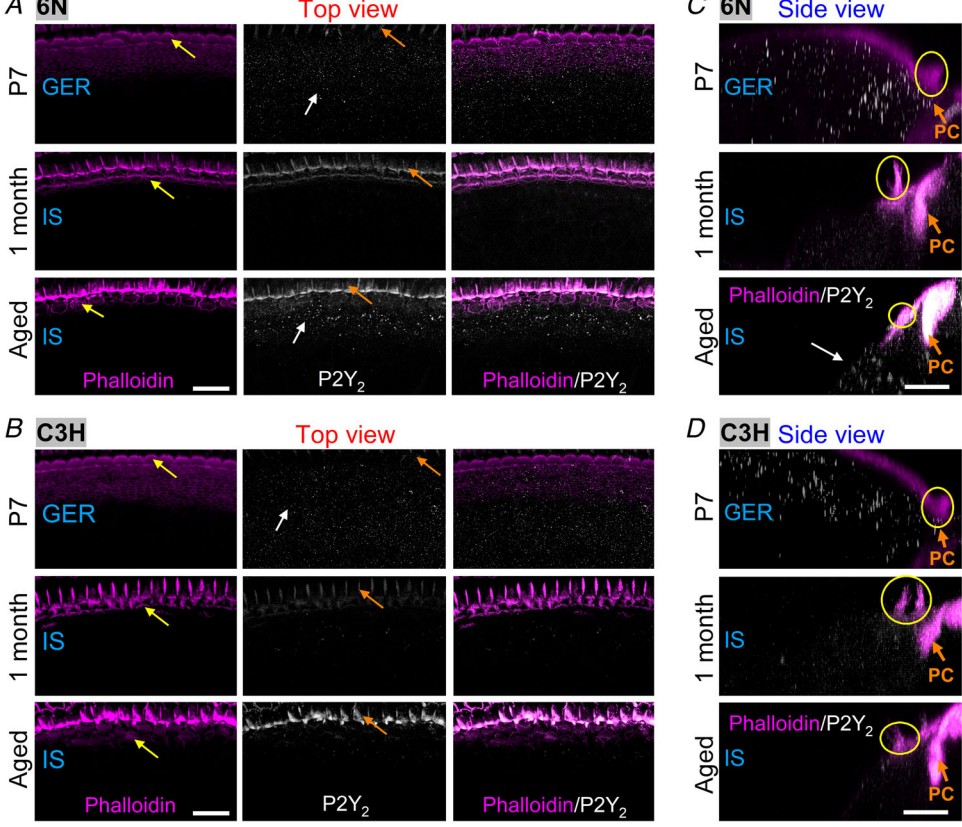

**Figure 3. Expression of P2Y$_2$ receptors in the ageing cochlea from 6N and C3H mice**
*A* and *B*, maximum intensity projections of confocal *z*-stacks showing images of the GER and IS viewed from the top (red arrow in panel 2*A*) in P7 (top panels), 1-month-old (middle panels) and aged (17–21 months: bottom panels) 6N (*A*) and C3H (*B*) mice. Left column: actin-marker phalloidin (magenta); middle column: P2Y$_2$ (white). Right panels show the merged images. For the identification of the cellular organisation and labels, see Fig. 2. Scale bars are 20 $\mu$m. *C* and *D*, maximum intensity projections of confocal *z*-stack images of the GER and IS viewed from the side (see Fig. 2*A*). Phalloidin: magenta; P2Y$_4$: white. Scale bars are 10 $\mu$m.

and down-regulation of P2Y receptor expression in the inner sulcus after the onset of hearing (Huang et al., 2010). Although ATP-induced $Ca^{2+}$ responses did not change in size between 1 and 6 months of age (average: $P = 0.1246$, maximum: $P > 0.9999$), by 12 months of age they started to increase (1 *vs.* 12 months, $P < 0.0001$ for both average and maximum) and by 18–24 months resembled those recorded during pre-hearing stages (Fig. 6*O* and *P*).

To determine whether the increase in ATP sensitivity was due to the hearing loss phenotype of 6N mice, or a more general ageing phenomenon in the supporting cells, we performed $Ca^{2+}$ imaging experiments in C3H mice, which at 20 months of age still exhibit good hearing thresholds (Fig. 1). We found that cochlear supporting cells from mature C3H mice showed a very small age-related increase in the size of ATP-induced $Ca^{2+}$ responses (Fig. 6*F–I, N*), which did not significantly increase after 12 months of age ($P > 0.9999$ for both

average and maximum, pairwise Wilcoxon rank sum test, ART two-way ANOVA, Fig. 6*O* and *P*). Strain comparison showed that both the average and maximum $Ca^{2+}$ responses recorded from supporting cells were significantly higher in 6N compared to C3H mice at 18–24 months of age ($P < 0.0001$ for both comparisons), but not at P7–P8 ($P = 0.1647$ and $P > 0.9999$, respectively) (Fig. 6*N* and *O*). These data suggest that the increase in ATP sensitivity is linked to the more severe hearing loss phenotype of the 6N strain compared to the C3H, indicating a possible role in the progression of age-related hearing loss.

To further corroborate this conclusion, and to find whether the observed phenotype was linked to $Cdh23^{ahl}$ allele, we performed $Ca^{2+}$ imaging experiments in 6N-Repaired mice, which, despite their better over-all hearing sensitivity, share a similar low-frequency hearing loss with the co-isogenic 6N strain (Jeng, Ceriani

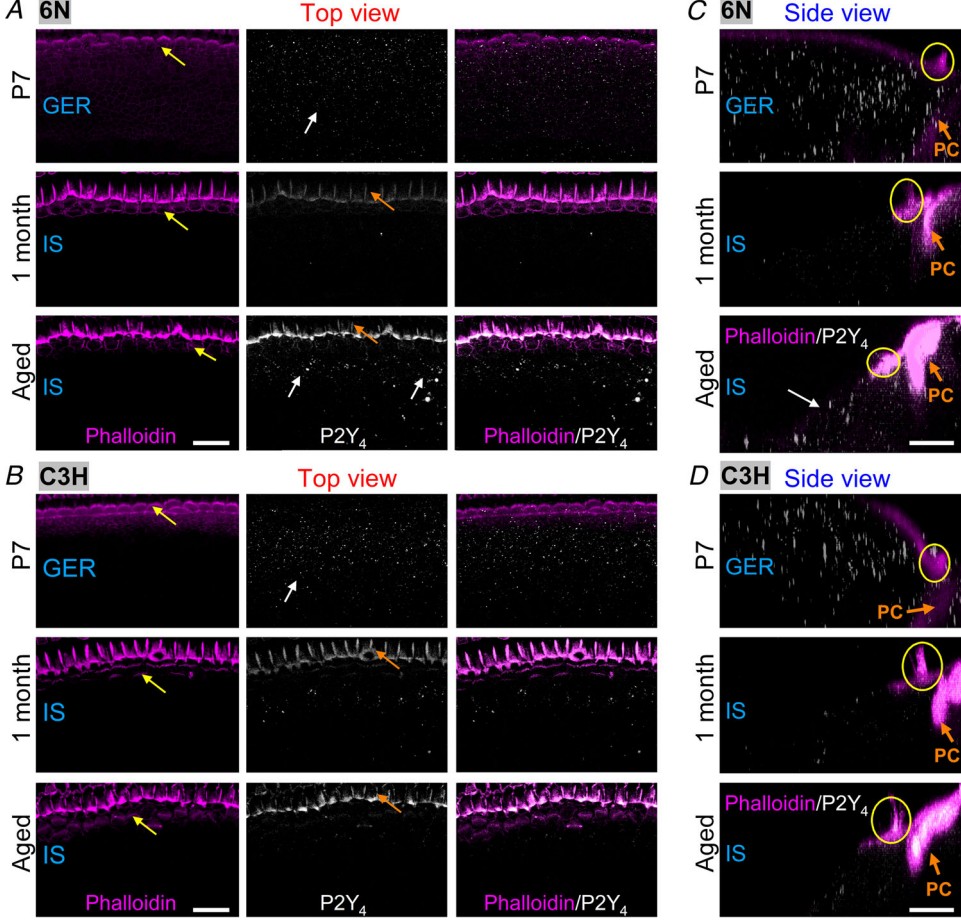

**Figure 4. Expression of P2Y$_4$ receptors in the ageing cochlea from 6N and C3H mice**
*A* and *B*, maximum intensity projections of confocal *z*-stacks showing images of the GER and IS viewed from the top (red arrow in panel 2*A*) in P7 (top panels), 1-month-old (middle panels) and aged (17–21 months: bottom panels) 6N (*A*) and C3H (*B*) mice. Left column: actin-marker phalloidin (magenta); middle column: P2Y$_4$ (white). Right panels show the merged images. For the identification of the cellular organization and labels, see Fig. 2. Scale bars are 20 $\mu$m. *C* and *D*, maximum intensity projections of confocal *z*-stack images of the GER and IS viewed from the side (see Fig. 2*A*). Phalloidin: magenta; P2Y$_4$: white. Scale bars are 10 $\mu$m.

et al., 2020). In 6N-Repaired mice, ATP-induced $Ca^{2+}$ responses in the supporting cells also increased with age, with a similar progression as the one observed in 6N mice (Fig. 6*J–N*) since no significant difference was found when comparing both their average and maximum $Ca^{2+}$ responses ($P = 0.7176$ and $P > 0.9999$, respectively: Tukey's *post hoc* test, ART two-way ANOVA, Fig. 6*O* and *P*). Beside the increase in size, a prominent feature of $Ca^{2+}$ responses to prolonged ATP application was the presence of both sustained $Ca^{2+}$ oscillations and repetitive $Ca^{2+}$ spikes in several supporting cells. The frequency of these $Ca^{2+}$ oscillations in the supporting cells increased with age in both 6N and 6N-Repaired adult mice (Fig. 6*Q*). Conversely, this increase in oscillatory behaviour was much less prominent in the C3H strain (Fig. 6*Q*).

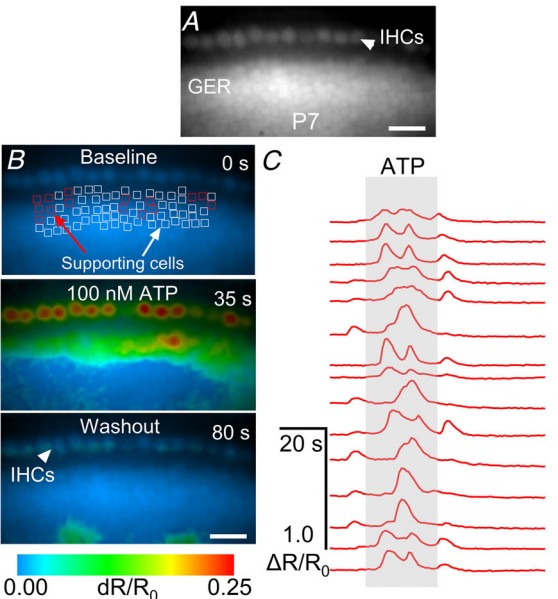

**Figure 5. Calcium imaging in the supporting cells of the immature mouse cochlea during ATP application**
A, fluorescence image showing the apical coil of the cochlea loaded with Fura-2 $Ca^{2+}$ dye. Fluorescence was excited at 385 nm. The arrowhead shows the location of the IHCs. Scale bar is 20 $\mu$m. B, false-colour images before (top), during (middle) and after (bottom) the application of 100 nM ATP. Images show the ATP-dependent $Ca^{2+}$ responses in supporting cells of the pre-hearing cochlea (P7) of a 6N mouse loaded with Fura-2 $Ca^{2+}$ dye. Square regions of interest (ROIs, red and white) were manually drawn on supporting cells in the proximity of the IHCs. These regions include inner phalangeal cells and inner border cells. The arrowhead in the bottom panel shows the location of the IHCs. Note that immature IHCs also exhibit spontaneous $Ca^{2+}$ responses, which are known to be triggered by ATP release from supporting cells (e.g. Carlton et al., 2023; Johnson et al., 2017; Tritsch et al., 2007). Scale bar is 20 $\mu$m. C, $Ca^{2+}$ traces, calculated as fractional change of the ratio between the fluorescence emission for excitation at 365 nm and 385 nm ($\Delta R/R_0$), respectively, for a subset of ROIs (red ROIs shown in panel *B*). ATP (100 nM) was applied during the time-window highlighted in grey.

To measure the sensitivity of $Ca^{2+}$ responses to extracellular ATP, we applied different concentrations of the purinergic agonist onto the supporting cells of 18- to 24-month-old 6N mice (Fig. 7). Calcium oscillations were visible in supporting cells in the presence of 30–300 nM ATP, while $Ca^{2+}$ responses tended to plateau after an initial peak at a higher concentration (1 $\mu$m, Fig. 7*A* and *B*). The concentration of ATP able to trigger 50% of the average and maximum $Ca^{2+}$ responses was around 60–70 nM (Fig. 7*C* and *D*), which is reasonably close to that previously reported for the supporting cells of the LER of pre-hearing mice (23 nM, Gale et al., 2004) and that present in extracellular cochlear fluids at rest (10–20 nM, Muñoz et al., 1995; Muñoz et al., 2001).

### ATP-induced $Ca^{2+}$ responses depend on $Ca^{2+}$ released from intracellular stores

In the immature cochlea, ATP-dependent activation of G-coupled P2Y receptors promotes the release of $Ca^{2+}$ from intracellular stores (Ceriani et al., 2016; Gale et al., 2004; Piazza et al., 2007). We therefore tested whether ATP-induced $Ca^{2+}$ signals in supporting cells from the aged cochlea depend on a similar mechanism. Thapsigargin is a known inhibitor of sarcoplasmic/endoplasmic reticulum $Ca^{2+}$-ATPase (SERCA) (Thastrup, 1990), thus preventing the refilling of $Ca^{2+}$ into the stores. We found that incubation of the cochlear preparation with 2 $\mu$m thapsigargin completely abolished the ATP-dependent $Ca^{2+}$ responses in the supporting cells of aged mice ($P < 0.0001$, Wilcoxon signed-rank test, Fig. 8*A* and *B*), as also observed in the pre-hearing cochlea ($P < 0.0001$, Fig. 8*B*, see also Babola et al., 2020; Piazza et al., 2007). This suggests that the $Ca^{2+}$ responses observed in the aged cochlea using sub-micromolar concentrations of ATP are mainly due to the activation of metabotropic purinergic (P2Y) receptors functionally linked to intracellular $Ca^{2+}$ stores.

### Calcium responses mediated by P2Y$_1$ receptors do not change with age

Having established that P2Y are the primary purinergic receptors underlying the increase in $Ca^{2+}$ responses in the supporting cells of the aged cochlea, we determined whether all three P2Y receptor subtypes identified in the pre-hearing cochlea (P2Y$_2$ and P2Y$_4$: Huang et al., 2010; Piazza et al., 2007; P2Y$_1$: Babola et al., 2020; see also Figs 2–4) were involved.

To investigate the possible contribution of P2Y$_1$ receptors, we used adenosine 5′-diphosphate (ADP), which is a known agonist for the P2Y$_1$, P2Y$_{12}$ and P2Y$_{13}$ receptors (Waldo et al., 2004, von Kügelgen et al., 2016). We found that 1 $\mu$m ADP elicited large $Ca^{2+}$ increases

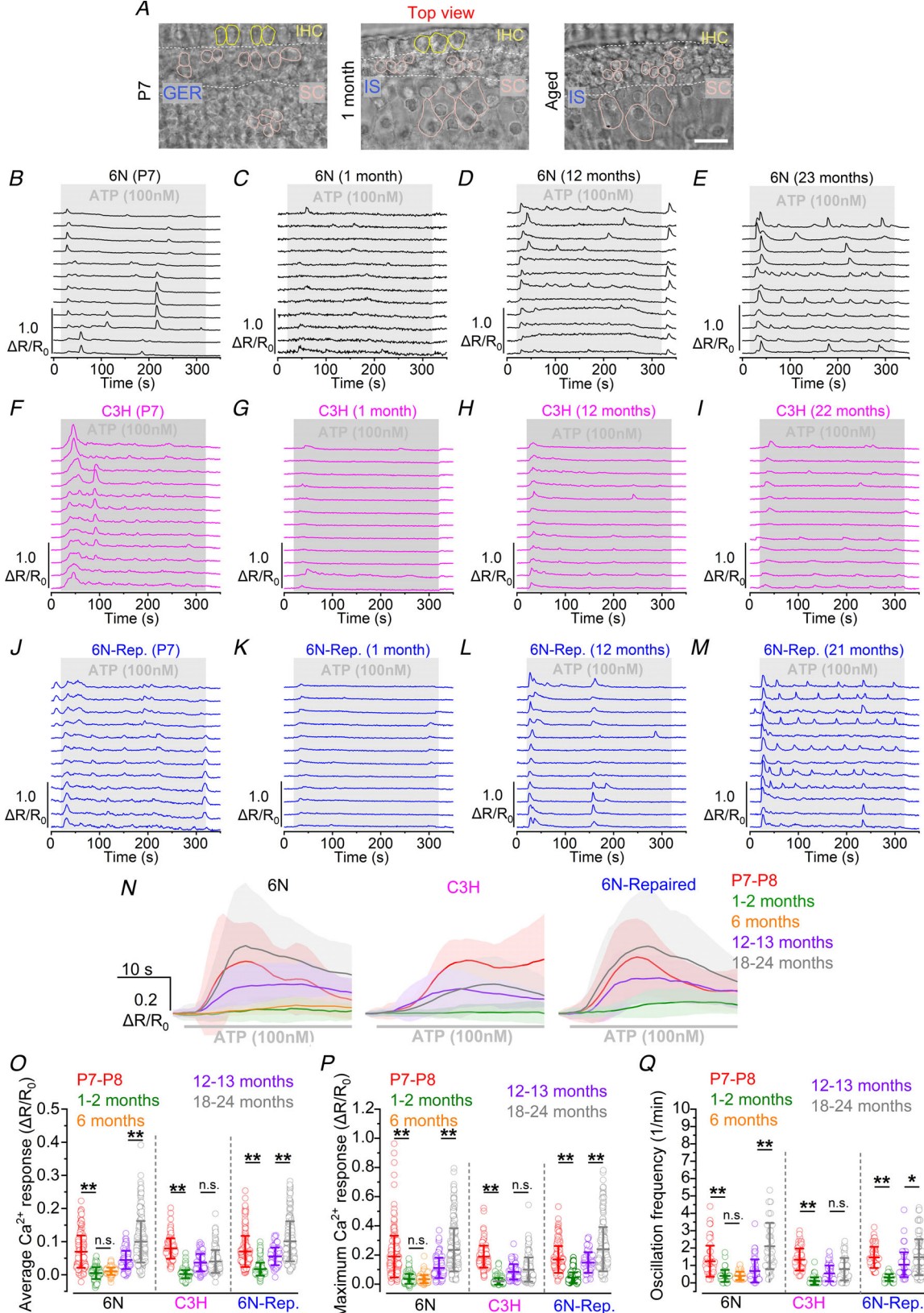

**Figure 6. ATP-induced Ca²⁺ responses in ageing 6N, C3H and 6N-Repaired ageing mice**

*A*, brightfield images of the organ of Corti in P7 (left), 1-month-old (middle) and aged (right) mice from Fig. 2*B* (immunolabelling experiments) showing the area of the GER and IS used for the Ca²⁺ imaging experiments, which includes the supporting cells close to the IHCs (area in between the white dashed lines). Scale bars are 20 μm. *B–M*,

representative Ca$^{2+}$ responses in supporting cells induced by the extracellular application of 100 nM ATP (grey area) in 6N (*B–E*, black), C3H (*F–I*, magenta), and 6N-Repaired (6N-Rep.: *J–M*, blue) mice at different ages reported above each set of traces. Synchronised peaks in several traces after the increase at the onset of stimulation (evident in panels *B*, *F* and *L*) reflect the propagation of Ca$^{2+}$ waves across multiple cells. *N*, comparison of the average Ca$^{2+}$ responses from supporting cells at the onset of ATP application from 6N (left), C3H (middle), and 6N-Repaired (right) mice at different ages. Continuous traces represent averages, while the shaded area is the SD. Numbers of individual ROIs (i.e. supporting cells) for mice at P7–P8, 1–2 months, 12–13 months, and 18–24 months are: 6N, 212 (8 mice), 144 (7), 134 (3), 276 (10); C3H, 127 (3), 194 (7), 127 (4), 173 (9); Repaired, 176 (6), 171 (5), 88 (3), 250 (8). For 6N mice, an additional set of experiments was performed in 6-month-old animals (130 ROIs from 4 mice). *O* and *P*, comparison of the average (*O*) and maximum (*P*) Ca$^{2+}$ response to 100 nM ATP application in the three mouse strains at different ages. Open circles represent single data points. Significance values are indicated by the asterisks (\*\**P* < 0.0001, Wilcoxon rank sum test, ART two-way ANOVA). *Q*, comparison of the oscillation frequency in supporting cells during application of 100 nM ATP. For this quantification, we only used experiments from panels *O* and *P* in which ATP perfusion was longer than 100 s. Numbers of individual ROIs (i.e. supporting cells) for mice at P7–P8, 1–2 months, 12–13 months, and 18–24 months are: 6N, 60 (3 mice), 52 (3), 55 (2), 48 (3); C3H, 66 (3), 71 (3), 65 (4), 56 (5); Repaired, 66 (3), 65 (4), 77 (3), 132 (6). For 6N mice, an additional set of experiments was performed in 6-month-old animals (55 ROIs from 4 mice). Open circles represent single data points. Significance values are indicated by the asterisks (\**P* = 0.031, \*\**P* < 0.0001, Wilcoxon rank sum test, ART two-way ANOVA).

in the supporting cells of 6N mice at all ages tested, including in the young adult cochlea (Fig. 9*A–C*), an age when they could not be triggered by the application of physiological concentrations of ATP (Fig. 6). Both the average and maximum of the Ca$^{2+}$ responses measured from supporting cells in the presence of ADP were found to be not significantly different between young adult (1–2 months) and aged mice (average Ca$^{2+}$ response:

$P$ = 0.6912; maximum Ca$^{2+}$ response: $P$ = 0.9999, Wilcoxon rank sum test, Kruskal Wallis test, Fig. 9*D–F*). The extracellular application of the specific P2Y$_1$ receptor antagonist MRS2500 (Hechler et al., 2006) fully blocked ADP-induced Ca$^{2+}$ responses, further confirming that P2Y$_1$ receptors underlie these (maximum Ca$^{2+}$ response: $P$ < 0.0001, Wilcoxon signed-rank test, Fig. 9*G* and *H*). Overall, as the responses to ADP showed no age-related

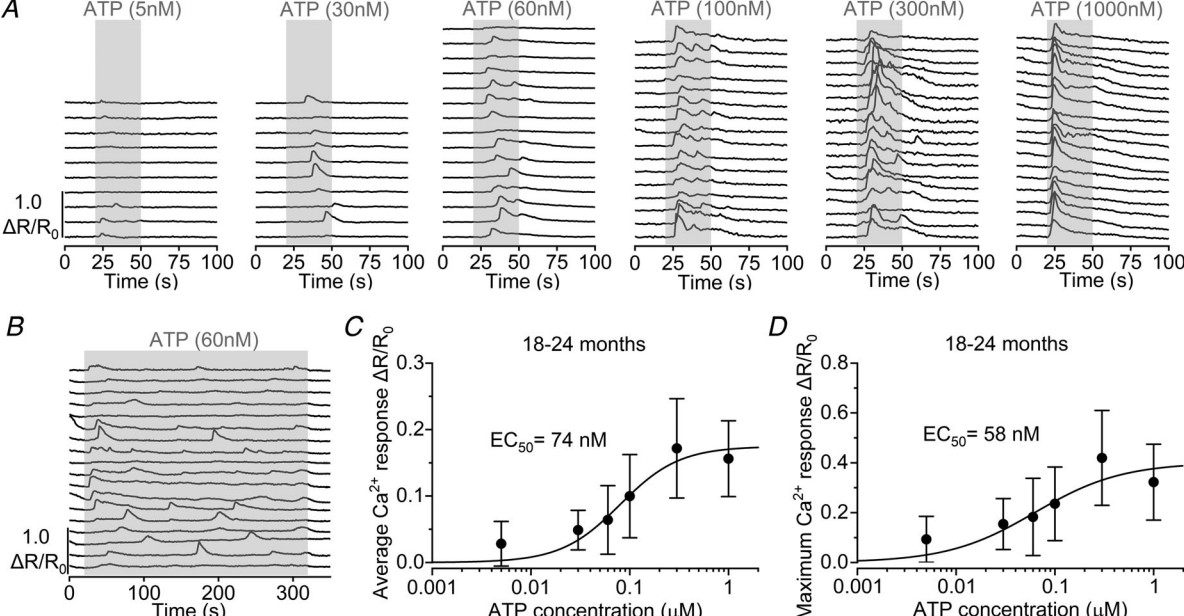

**Figure 7. Dose-dependent ATP-induced Ca$^{2+}$ signalling in supporting cells from aged 6N mice**
*A*, representative Ca$^{2+}$ responses to different concentrations of ATP in supporting cells induced by ATP perfusion (grey area) in aged (18–24 months) 6N mice. *B*, prolonged stimulation (5 min) with 60 nM ATP, highlighting the occurrence of slow oscillations. *C* and *D*, dose-response curves for the average (*C*) and maximal (*D*) Ca$^{2+}$ response as a function of the ATP concentration. Data are plotted as mean ± SD. The continuous lines represent a fit with a Hill function $R = R_{max} \frac{[ATP]^n}{[ATP]^n + EC_{50}^n}$, with EC$_{50}$ = 74 ± 6 nM and $n$ = 1.62 ± 0.23 (*C*) and EC$_{50}$ = 58 ± 10 nM and $n$ = 1.07 ± 0.22 (*D*). Numbers of individual supporting cells (ROIs) recorded from 18–24-month-old mice and from lower to higher concentrations are: 31 (3 mice), 56 (3), 130 (6), 276 (10), 66 (3), 56 (5).

relationship after the onset of hearing, these results indicate that $P2Y_1$ receptors do not contribute to the re-appearance of ATP-induced $Ca^{2+}$ responses seen in the aged cochlea.

### $P2Y_2$- and $P2Y_4$-mediated $Ca^{2+}$ responses increase in aged mice

The application of 300 nM UTP, a selective agonist of $P2Y_2$ and $P2Y_4$ receptors that mobilises $Ca^{2+}$ from intracellular stores (Piazza et al., 2007), triggered large $Ca^{2+}$ responses in both pre-hearing and aged supporting cells from the 6N mouse strain, with some cells displaying prominent $Ca^{2+}$ oscillations (Fig. 10A–C). These UTP-induced responses followed a similar age-dependent time course as those induced by ATP (Fig. 6O and P), with a reduction in amplitude at 1–2 months of age and a subsequent increase in 18- to 24-month-old mice. A similar effect was also observed in ageing C3H mice (Fig. 10D–F), although UTP-induced $Ca^{2+}$ responses in 18- to 24-month-old mice were much smaller compared to age-matched 6N (average and maximum $Ca^{2+}$ response: $P < 0.0001$, pairwise Wilcoxon rank-sum test, ART two-way ANOVA, Fig. 10G–I). These results confirm that purinergic-induced $Ca^{2+}$ signals also reappear in the ageing cochlea from C3H mice, albeit at later ages compared to the 6N strain, as also suggested when ATP was used (Fig. 6). To investigate whether both $P2Y_2$ and $P2Y_4$ were involved in the UTP-induced $Ca^{2+}$ signals in the aged cochlea, we performed some experiments using the selective $P2Y_2$ antagonist AR-C 118925XX (Rafehi et al., 2017) and $P2Y_4$ agonist MRS4062 (Maruoka et al., 2011). We found that AR-C 118925XX abolished the

UTP-induced $Ca^{2+}$ responses (Fig. 11A and B), while MRS4062 produced similar $Ca^{2+}$ signals to those elicited by UTP (Fig. 11C and D). These findings confirm the involvement of both $P2Y_2$ and $P2Y_4$ receptors in the generation of $Ca^{2+}$ responses in the supporting cells of the aged cochlea.

## Discussion

Purinergic signalling has been shown to play a key role not only in the development and function of the mammalian cochlea, but also in cochlear pathophysiology (Vlajkovic & Thorne, 2022). In this study, we investigated responses to nanomolar ATP application, a concentration that has been shown to mainly activate $Ca^{2+}$-mobilising metabotropic purinergic receptors (P2Y) in the cochlea (Ceriani et al., 2016; Gale et al., 2004; Piazza et al., 2007). We found that the expression of $P2Y_1$, $P2Y_2$ and $P2Y_4$ receptors and associated ATP-induced $Ca^{2+}$ responses in the cochlear supporting cells are down-regulated following the onset of hearing, but then increase in the aged cochlea of mice older than 6 months. Although the upregulation of ATP-induced $Ca^{2+}$ signalling was a general feature of the ageing cochlea, it was much more pronounced in mouse models with early onset low-frequency hearing loss (C57BL/6N) compared to mice with preserved hearing sensitivity (C3H-HeJ). We also showed a similar age-related increase in $Ca^{2+}$ responses with UTP ($P2Y_2$ and $P2Y_4$ agonist), but not ADP ($P2Y_1$ agonist), supporting the upregulation of $P2Y_2$ and $P2Y_4$ receptors in the aged cochlea observed in our immunolabelling experiments. These ATP-induced $Ca^{2+}$ responses in the supporting cells of the aged cochlea resemble those appearing spontaneously in the developing cochlea, which are required for normal development of the auditory pathway (Babola et al., 2020; Tritsch & Bergles 2010), including the hair cells (Johnson et al., 2017). Note that the upregulation of P2Y-mediated $Ca^{2+}$ signalling in the supporting cells occurred after the appearance of age-related changes in the hair cells (at 6 months: Jeng et al., 2021; Jeng, Johnson et al., 2020). It would be interesting in the future to understand whether these changes are simply due to a progressive deterioration of the mammalian cochlea with age, or an attempt to 'recapitulate' early developmental signalling, possibly as part of attempts to repair the malfunctioning hair cells.

### Metabotropic purinergic receptors in the supporting cells of the developing, adult and aged mouse cochlear inner sulcus

Metabotropic P2Y receptors show widespread expression in the cochlea, both in sensory and supporting cells

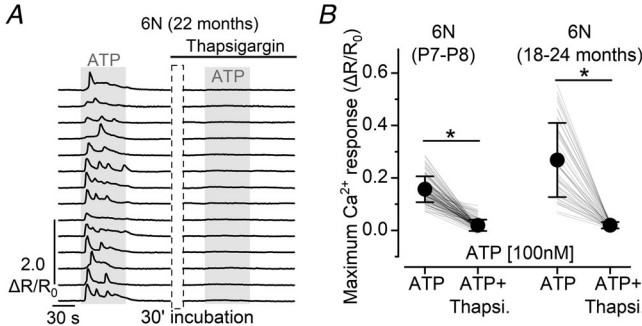

**Figure 8. ATP-induced $Ca^{2+}$ signals in supporting cells of aged mice depend on intracellular $Ca^{2+}$ stores**
A, representative $Ca^{2+}$ responses in supporting cells induced by ATP perfusion (100 nM, grey area) in an aged 6N mouse before (left) and after (right) 30-min incubation in 2 $\mu$m thapsigargin. B, maximal ATP-induced $Ca^{2+}$ response before and after thapsigargin incubation. Thapsigargin incubation abolished the $Ca^{2+}$ responses ($P < 0.0001$, Wilcoxon signed-rank test). Numbers of individual supporting cells (ROIs): P7–P8, 102 (3 mice); 18–24 months old, 46 (3).

(Huang et al., 2010; Köles et al., 2019). During pre-hearing stages of cochlear development, the P2Y receptors expressed in the supporting cells of the GER (Babola et al., 2021; Huang et al., 2010) play a crucial role in the propagation of slow spontaneous intercellular $Ca^{2+}$ waves across the immature epithelium (Babola et al., 2020; Johnson et al., 2017; Tritsch & Bergles, 2010; Tritsch et al.,

2007). This ATP-dependent $Ca^{2+}$ signalling is mediated by P2Y receptors and has been shown to regulate not only the development and fine tuning of sensory hair cells and their innervation (Ceriani et al., 2019; Johnson et al., 2017), but also the refinement of tonotopic maps in the brain (Babola et al., 2018; Babola et al., 2021; Kersbergen et al., 2022). After the onset of hearing, the cochlea reaches

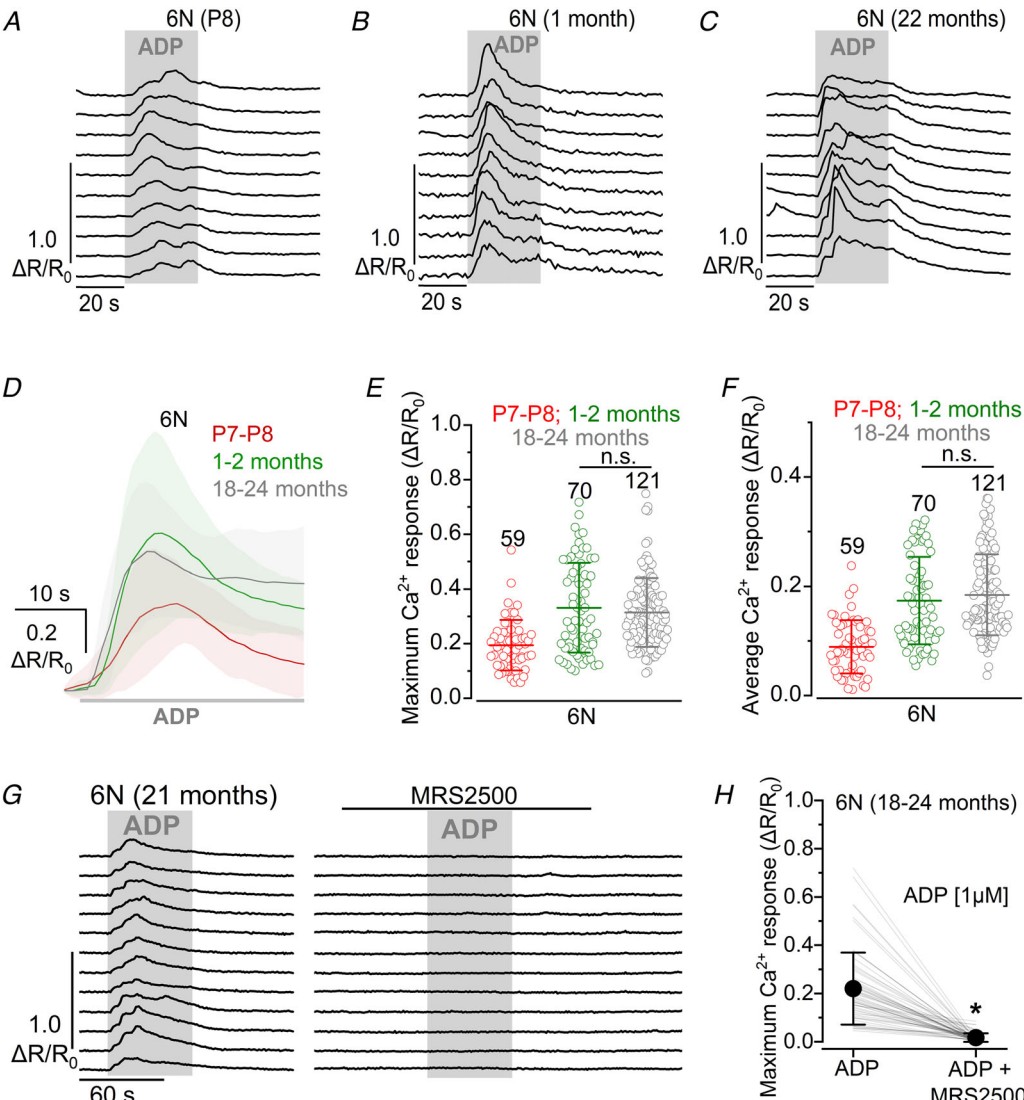

**Figure 9. ADP-induced $Ca^{2+}$ responses in supporting cells of the 6N mouse cochlea**
*A–C*, representative $Ca^{2+}$ responses in supporting cells induced by the extracellular application of 1 $\mu$m ADP (grey area) in 6N mice at different age ranges shown above the recordings. *D*, comparison of the average $Ca^{2+}$ response at the onset of ADP application (grey bar beneath the traces) in cochlear supporting cells of 6N mice in the three different age ranges tested. Continuous traces represent averages, while the shaded area is the SD. Numbers of individual supporting cells (ROIs): P7–P8, 59 ROIs (3 mice); 1–2 months old, 70 ROIs (4), and 18–24 months old, 121 ROIs (5). *E* and *F*, comparison of the maximum (*E*) and average (*F*) $Ca^{2+}$ response to 1 $\mu$m ADP application in 6N mice at different ages. Number of supporting cells used is shown above the averages ($\pm$ SD) and single data points (plotted as open circles). *G*, representative $Ca^{2+}$ responses in supporting cells induced by 1 $\mu$m extracellular ADP (grey area) in aged 6N mice. The application of ADP together with the P2Y$_1$ antagonist MRS2500 (1 $\mu$m, top black horizontal line) blocked the ADP-induced $Ca^{2+}$ response. *H*, effect of P2Y$_1$ antagonist MRS2500 on the size of the ADP-induced $Ca^{2+}$ response in cochlear supporting cells from aged mice from 64 supporting cells (ROIs) from 3 mice. Significance values are indicated by the asterisks ($P < 0.0001$, Wilcoxon signed-rank test).

functional maturity, and the inner sulcus (IS) replaces the GER. Supporting cells in the IS of the mouse cochlea no longer elicit the spontaneous $Ca^{2+}$ waves and the slow ATP-induced currents characteristic of P2Y receptor activation (Sirko et al., 2019). Indeed, previous work has shown that in rats the expression of P2Y receptors in the supporting cells of the inner sulcus is reduced compared to that in the immature GER (Huang et al., 2010).

Previous investigations have shown that P2Y-mediated $Ca^{2+}$ responses in cochlear supporting cells are elicited by micromolar concentrations of ATP (e.g. Horvath et al., 2016; Lahne & Gale, 2008; Rabbitt & Holman,

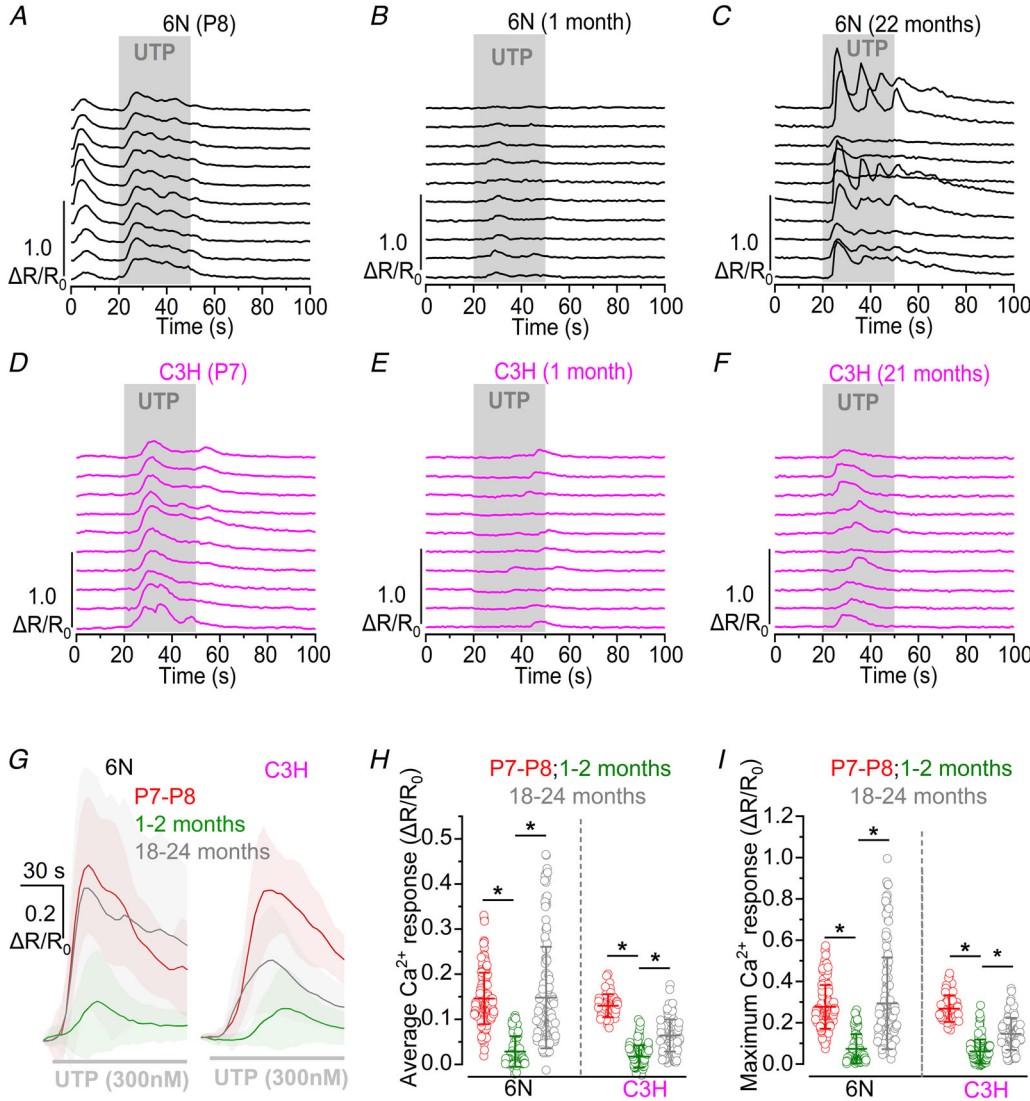

**Figure 10. Age dependence of UTP-induced $Ca^{2+}$ responses in ageing mice**
*A–F*, representative $Ca^{2+}$ responses in supporting cells induced by the superfusion of 300 nM UTP (grey area) in 6N (*A–C*, black) and C3H (*D–F*, magenta) mice at the different age ranges shown above the recordings. *G*, comparison of the average $Ca^{2+}$ response at the onset of UTP application (grey bar beneath the traces) in cochlear supporting cells of 6N (left) and C3H (right) mice. Continuous traces represent averages, while the shaded area is the SD. Numbers of individual supporting cells (ROIs) for P7–P8, 1- to 2-month-old and 18- to 24-month-old mice, are: 6N, 151 (5 mice), 104 (5), 148 (7); C3H, 66 (3), 128 (5), 125 (7). The average frequency of UTP-induced $Ca^{2+}$ oscillations for 6N mice at different ages was: P7–P8, 3.75 ± 1.29 oscillations/min; 1–2 months, 1.33 ± 1.29 oscillations/min; 18–24 months, 3.83 ± 2.35 oscillations/min. The average frequency of UTP-induced $Ca^{2+}$ oscillations for C3H mice at different ages was: P7–P8, 3.32 ± 1.24 oscillations/min; 1–2 months, 1.24 ± 1.05 oscillations/min; 18–24 months, 1.57 ± 0.85 oscillations/min. *H* and *I*, comparison of the average (*H*) and maximum (*I*) $Ca^{2+}$ response to 300 nM UTP application in 6N and C3H mice at different ages. Single data points are plotted as open circles. Numbers of individual supporting cells (ROIs) and mice is as described in panel *G* above. Significance values are indicated by the asterisks. *$P < 0.0001$, Wilcoxon rank sum test, ART two-way ANOVA.

2021; Sirko et al., 2019; Tritsch & Bergles 2010). We showed that $Ca^{2+}$ responses can also be elicited by nanomolar ATP, which approaches the concentration present in extracellular cochlear fluids *in vivo* (Muñoz et al., 1995; Muñoz et al., 2001). Although the contribution of P2Y receptors to ATP-mediated $Ca^{2+}$ responses is well established, the nature of the receptor subtype is still debated. Classical studies have provided evidence showing the involvement of $P2Y_2$ and $P2Y_4$ receptors (Huang et al., 2010; Piazza et al., 2007). However, more recent findings have shown that $P2Y_1$ receptors are the main subtype

driving purinergic-mediated spontaneous $Ca^{2+}$ responses in the supporting cells of the GER in the prehearing cochlea (Babola et al., 2021).

Our data show that $Ca^{2+}$ responses in the supporting cells nearby the IHCs of the pre-hearing cochlea can be elicited not only by the $P2Y_1$ agonist ADP, but also by the $P2Y_2$ and $P2Y_4$ receptor agonist UTP. This finding indicates that all three purinergic subtypes may contribute to the $Ca^{2+}$ responses in the supporting cells of the pre-hearing cochlea (see also Babola et al., 2020; Huang et al., 2010). Since $P2Y_1$ affinity for ADP is 100 times that of ATP (von Kügelgen & Hoffmann, 2016), it is possible that ATP in the prehearing cochlea is readily converted into ADP by ectonucleotidases (O'Keefe et al., 2010). Alternatively, several members of the P2Y receptor family could combine to form hetero-oligomers, altering the receptors properties, including their pharmacology and trafficking (Ecke et al., 2008; Nakata et al., 2010). It is thus possible that interactions between the different P2Y receptor subtypes could influence their response to agonists/antagonists. Although our $Ca^{2+}$ imaging approach combined with purinergic pharmacology directly implicates $P2Y_2$ and $P2Y_4$ in the reappearance of ATP-induced $Ca^{2+}$ signalling in the aged cochlea, it is very likely that other mechanisms such as cytoplasmic $Ca^{2+}$ buffering or $Ca^{2+}$ clearance may change with age. Although very little is known about these additional mechanisms in the aged mouse cochlea, a dysregulation of neuronal $Ca^{2+}$ dynamics has previously been associated with ageing of the nervous system (Mattson et al., 2018). Moreover, it remains to be determined whether other mechanisms known to be involved in the propagation of intercellular $Ca^{2+}$ responses in the immature cochlea, such as diffusion of second messengers (e.g. $IP_3$) between supporting cells through gap junction channels (Anselmi et al., 2008; Ceriani et al., 2016), are also changed in the ageing cochlea.

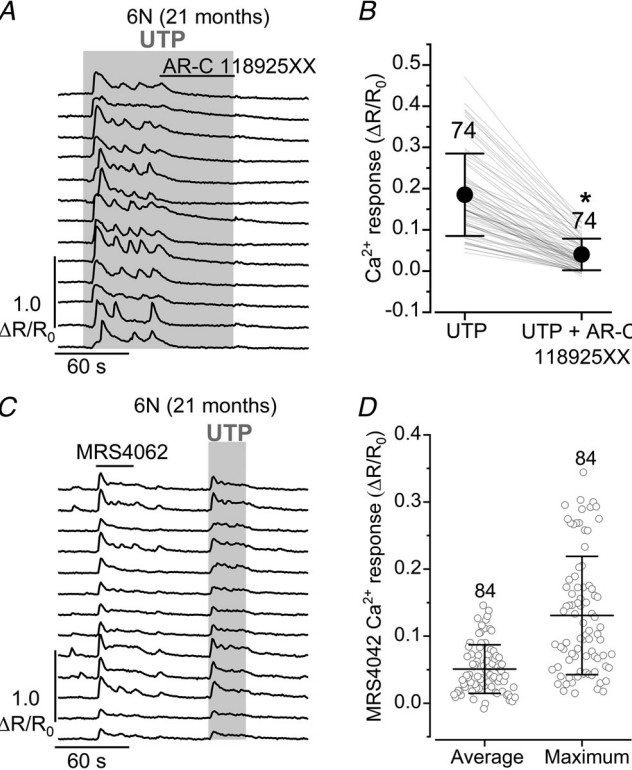

**Figure 11. Pharmacology of UTP-induced $Ca^{2+}$ responses in supporting cells**

*A*, representative $Ca^{2+}$ responses in supporting cells induced by the extracellular application of 300 nM UTP (grey area) in aged 6N mice. The application of UTP together with the $P2Y_2$ antagonist ARC-118925XX (15 μM, top black horizontal line) caused a reduction of the UTP-induced $Ca^{2+}$ response and stopped $Ca^{2+}$ oscillations. *B*, average UTP-induced $Ca^{2+}$ responses during the last 10 s of UTP application and when UTP was applied together with the $P2Y_2$ antagonist ARC-118925XX in cochlear supporting cells from aged mice (74 ROIs from 4 mice). Significance values are indicated by the asterisks ($P < 0.0001$, Wilcoxon signed-rank test). *C*, representative $Ca^{2+}$ responses in supporting cells induced by the $P2Y_4$ agonist MRS4062 (10 μM, top black horizontal line) in aged 6N mice. This initial response was followed by the application of 300 nM UTP alone (grey area), to confirm that the supporting cell was responsive to UTP. *D*, average and maximum MRS4062-induced $Ca^{2+}$ response in cochlear supporting cells from aged mice. Open symbols are measurements from individual supporting cells (ROIs): 84 from 3 mice.

## Role of purinergic receptors in the cochlea

Several lines of evidence indicate that one of the most common causes of age-related hearing loss in mice is the change in the morphology and function of the sensory hair cells and their innervation within the cochlea (Lauer et al., 2012; Liberman, 2017; Jeng et al., 2021; Jeng, Ceriani et al., 2020; Jeng, Johnson et al., 2020). In this study we showed that by 12 months of age, the supporting cells located in the low-frequency cochlear region (9–12 kHz) of 6N and the co-isogenic strain 6N-Repaired mice upregulate $P2Y_2$ and $P2Y_4$ receptors. Both 6N and 6N-Repaired mice exhibit a comparable progressive low-frequency hearing loss for frequencies below 18 kHz (Jeng, Ceriani et al., 2020). Different from the 6N and 6N-Repaired mice, the C3H strain retains

very good hearing thresholds across the frequency range even at older ages, and their hair cells and associated innervation have normal structure and function until at least 15–18 months of age (Jeng et al., 2021; Jeng, Ceriani et al., 2020; Jeng, Johnson et al., 2020). The supporting cells present in the cochlea of C3H mice showed minimal age-related changes in the expression of P2Y receptors and mediated $Ca^{2+}$ signalling. These findings indicate that, like hair cells (Jeng et al., 2021; Jeng, Ceriani et al., 2020; Jeng, Johnson et al., 2020), the supporting cells undergo specific functional changes during ageing that signify the level of progressive loss of auditory function.

Supporting cells can act as phagocytes by engulfing and removing dying hair cells (Anttonen et al., 2014; Monzack et al., 2015). Moreover, the release of ATP from the hair cells has previously been proposed as the 'eat me' signal that activates their removal by nearby supporting cells (Bird et al., 2010). Therefore, the increased activity of purinergic receptors in the aged cochlea could participate in control of hair cell death, for example during immune responses, which may be associated with age-related hearing loss (Köles et al., 2019; Verschuur et al., 2014). This mechanism, if present, could be driven by $P2Y_2$ receptors since they have been linked to immune cell migration to the site of inflammation, with ATP released from damaged cells acting as an attracting chemotactic signal (Chen et al., 2006). Interestingly, the ATP-induced $Ca^{2+}$ responses in the supporting cells of aged mice resemble those present in the developing cochlea (Piazza et al., 2007), in which ATP can be released following damage to hair cells (Gale et al., 2004; Lahne & Gale, 2008; Lahne & Gale, 2010; Piazza et al., 2007). However, there is very little or no loss of hair cells in the cochlear region investigated in this study (9–12 kHz), at least up to about 12 months of age (Jeng et al., 2021; Jeng, Johnson et al., 2020), a time when the purinergic signalling is already upregulated. Therefore, the involvement of the observed age-related changes in P2Y receptors in cochlear immune responses seems unlikely.

Changes in P2Y-mediated signalling in the aged cochlea could potentially act as a protective mechanism to limit or to avoid further damage caused by the several morphological and functional changes occurring in the sensory hair cells (e.g. Jeng et al., 2021; Jeng, Ceriani et al., 2020; Jeng, Johnson et al., 2020; Lauer et al., 2012), which may be perceived as cochlear insults. Indeed, it has been shown that the concentration of ATP in both endolymph and perilymph, and the induced $Ca^{2+}$ signals in the supporting cells, can increase following pathological insults such as noise, hypoxia, or ischaemia (Chan & Rouse, 2016; Muñoz et al., 1995; Muñoz et al., 2001). This increase as been interpreted as ATP having a protecting role on the cochlea during insults by mediating adaptation of cochlear responses (Housley et al., 2013) and regulating cochlear mechanics (Skellet et al., 1997).

As mentioned above, there are remarkable similarities between the ATP-induced $Ca^{2+}$ responses in the supporting cells of aged mice and those present in the developing cochlea (Babola et al., 2020; Johnson et al., 2017; Tritsch & Bergles 2010), which have been suggested to be involved, in addition to controling cell death, in modulating gene expression (Ortolano et al., 2008). A similar process in which the cochlear tissue seemingly 'recapitulates' the developmental configuration has been seen in the hair cells, with the efferent system re-forming direct axo-somatic contacts with the IHCs in aged mice (Jeng et al., 2021; Lauer et al., 2012). The upregulation of P2Y-mediated $Ca^{2+}$ signalling in the supporting cells occurred after some of the age-related changes in the hair cells, such as a reduction in their surface area and the size of $K^+$ currents, which were already evident at 6 months of age (Jeng et al., 2021; Jeng, Johnson et al., 2020). In the future, it will be interesting to understand whether the age-related changes in the efferent system and purinergic signalling are simply a consequence to the progressive deterioration of the mammalian cochlea with age, or instead an attempt to 'recapitulate' early developmental ages. The latter mechanisms, which are normally used to control the functional and structural remodelling of the maturing epithelium, could potentially be used to repair or limit the damage caused by the malfunctioning hair cells during ARHL.

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

## Additional information

### Data availability statement

The data that support the findings of this study are available from the corresponding author upon reasonable request.

### Competing interests

The authors declare no conflict of interest.

### Author contributions

All authors helped with the interpretation of the data and commenting on the manuscript. S.A.H. and F.C. performed the experiments. S.A.H., J.-Y. J., D.J., W.M., and F.C. contributed to the writing of the paper. W.M. and F.C. conceived and coordinated the study. All authors approved the final version of the manuscript. All authors agree to be accountable for all aspects of the work in ensuring that questions related to the accuracy or integrity of any part of the work are appropriately

investigated and resolved. All persons designated as authors qualify for authorship, and all those who qualify for authorship are listed.

## Funding

This work was supported by the BBSRC (BB/V006681/1 to F.C and W.M). S.A.H. is funded by a PhD studentship from the Royal National Institute for Deaf People (RNID; S58 to D.J and W.M.). J-Y.J. was supported by the RNID & Dunhill Medical Trust Fellowship (PA28).

## Acknowledgements

Confocal images were acquired using the Zeiss LSM 880 Airyscan microscope at the Wolfson Light Microscope Facility. The authors also thank Matthew A. Loczki for his assistance with the mouse colonies.

## Keywords

ageing, calcium signalling, cochlea, purinergic receptors, supporting cells

## Supporting information

Additional supporting information can be found online in the Supporting Information section at the end of the HTML view of the article. Supporting information files available:

**Statistical Summary Document**
**Peer Review History**

