## [Peer Review History · The Journal of Physiology]

Age-related changes in P2Y receptor signalling in mouse cochlear supporting cells

Sarah A Hool, Jing-Yi Jeng, Dan Jagger, Walter Marcotti, and Federico Ceriani
DOI: 10.1113/JP284980

Corresponding author(s): Walter Marcotti (w.marcotti@sheffield.ac.uk)

The following individual(s) involved in review of this submission have agreed to reveal their identity: Michael G Evans (Referee #2)

Review Timeline:

Submission Date:	04-May-2023
Editorial Decision:	07-Jun-2023
Revision Received:	12-Jul-2023
Editorial Decision:	07-Aug-2023
Revision Received:	14-Aug-2023
Accepted:	16-Aug-2023

Senior Editor: Katalin Toth

Reviewing Editor: Samuel Young

Transaction Report:

Dear Dr Ceriani,

Re: JP-RP-2023-284980 "Age-related changes in P2Y receptor signalling in the mouse cochlear supporting cells" by Sarah A Hool, Jing-Yi Jeng, Dan Jagger, Walter Marcotti, and Federico Ceriani

Thank you for submitting your manuscript to The Journal of Physiology. It has been assessed by a Reviewing Editor and by 2 expert referees and we are pleased to tell you that it is potentially acceptable for publication following satisfactory major revision.

REVISION CHECKLIST:

We look forward to receiving your revised submission.

Yours sincerely,

Katalin Toth
Senior Editor
The Journal of Physiology

REQUIRED ITEMS

-Author photo and profile. First (or joint first) authors are asked to provide a short biography (no more than 100 words for one author or 150 words in total for joint first authors) and a portrait photograph. These should be uploaded and clearly labelled with the revised version of the manuscript. See Information for Authors for further details.

-You must start the Methods section with a paragraph headed Ethical Approval. A detailed explanation of journal policy and regulations on animal experimentation is given in Principles and standards for reporting animal experiments in The Journal of Physiology and Experimental Physiology by David Grundy J Physiol, 593: 2547-2549. doi:10.1113/JP270818.). A checklist outlining these requirements and detailing the information that must be provided in the paper can be found at: <https://physoc.onlinelibrary.wiley.com/hub/animal-experiments>. Authors should confirm in their Methods section that their experiments were carried out according to the guidelines laid down by their institution's animal welfare committee, and conform to the principles and regulations as described in the Editorial by Grundy (2015). The Methods section must contain details of the anaesthetic regime: anaesthetic used, dose and route of administration and method of killing the experimental animals.

-Please upload separate high-quality figure files via the submission form.

-Please ensure that the Article File you upload is a Word file.

-A Statistical Summary Document, summarising the statistics presented in the manuscript, is required upon revision. It must be on the Journal's template, which can be downloaded from the link in the Statistical Summary Document section here: https://jp.msubmit.net/cgi-bin/main.plex?form_type=display_requirements#statistics

-Please include an Abstract Figure file, as well as the figure legend text within the main article file. The Abstract Figure is a piece of artwork designed to give readers an immediate understanding of the research and should summarise the main conclusions. If possible, the image should be easily 'readable' from left to right or top to bottom. It should show the physiological relevance of the manuscript so readers can assess the importance and content of its findings. Abstract Figures should not merely recapitulate other figures in the manuscript. Please try to keep the diagram as simple as possible and without superfluous information that may distract from the main conclusion(s). Abstract Figures must be provided by authors no later than the revised manuscript stage and should be uploaded as a separate file during online submission labelled as File Type 'Abstract Figure'. Please ensure that you include the figure legend in the main article file. All Abstract Figures should be created using BioRender. Authors should use The Journal's premium BioRender account to export high-resolution images. Details on how to use and access the premium account are included as part of this email.

EDITOR COMMENTS

Reviewing Editor:

This manuscript provides several new observations on the inner mechanisms that result in age-related hearing loss. Both reviewers found the data interesting, insightful and will be of broad interest to the hearing sciences research community. Both reviewers found much of the data very supportive of the conclusions. However, both reviewers have concerns with confocal imaging experiments as these data appear to be weakly supportive of conclusions drawn. To strengthen conclusions in the manuscript, the authors need to address the confocal imaging data concerns. Finally, the authors should carefully revise and rewrite their manuscript based on the positive criticism provided by the reviewers.

Ethics: Please add statement on unlimited access to food and water.

REFeree COMMENTS

Referee #1:

This manuscript by Hool et al. aims to describe the localization and functional properties of metabotropic purinergic P2Y receptors in supporting cells of the apical mouse cochlea with age. Cochlea tissue from mouse models showing early (6N) or late onset (C3H and 6N repaired) hearing loss were immunolabeled for purinergic receptors and compared. Ratiometric calcium imaging experiments in explants show supporting cells respond to extracellular application of ATP and purinergic receptor subtypes mediating Ca²⁺ responses are characterized with various agonists and antagonists. The authors find that expression of supporting cell P2Y1, P2Y2 and P2Y4 receptors decreases during postnatal cochlear maturation as described previously (consistent with the disappearance of Kolliker's organ). New findings show apparent upregulation of P2Y2 and P2Y4-mediated responses at older ages. The authors speculate that since the upregulation is greater in early onset hearing loss mice receptor changes may be protective to hair cells, although the mechanisms are not clear. Overall, the calcium imaging experiments with pharmacological characterization of purinergic responses in support cells of aging mice provide interesting new data. The conclusions from immunohistochemistry data are less well supported.

1. The cartoons showing distributions of cell types in immature and mature cochlea in Fig. 2 are helpful for orientation, but visualizing cell layers and arrangements in the confocal images is problematic. The authors should consider using cellular markers to label hair cells (anti-myosin 7A) and supporting cells (anti-Sox2). Immunolocalization of purinergic receptors in supporting cells of the medial compartment is unclear (Figs 2 and 3). Z-stack images (number of stacks and image acquisition details are not given) at different ages are shown and changes in receptor expression with age are claimed. But in some cases, receptor labeling looks the same as the actin labeling of hair bundles (e.g. P2Y2 signal and phalloidin signal in Fig. 3B). Also, phalloidin labelled hair bundles in Fig 2B middle image P7 are only partly visible. Since only representative images (with no quantification of immunofluorescence) are shown at the different ages, conclusions on age-related changes seem overstated.

2. As shown in Fig. 2A and described in Results (lines 263-271) there are several different types of supporting cell in the medial compartment region. Were attempts made to classify support cells by type when examining Ca²⁺ responses? In this regard it could be helpful if a brightfield image (mentioned on line 217) of the preparation could be shown along with the manually placed ROIs in Fig. 4A.

3. How was the "average Ca²⁺ response" measured (Fig. 5N)? Please describe the averaging technique in Methods. It's interesting that the maximum Ca²⁺ response was highly variable in the 6N mice at P7-8 (Fig. 5O), could you speculate why this might be? Also, a description of how "oscillation frequency" was measured for Fig. 5P would be helpful and describe how oscillations differ from repetitive Ca²⁺ spikes (lines 331-332).

4. The changes in P2Y signals with age are interesting, but their role in age-related hearing loss is unresolved. The Discussion of age-related changes (p17-18) is rather long and inconclusive. Suggest substantially shortening this section by deleting description of age-related changes in purinergic signaling in other systems. Ca²⁺ responses shown here are indirect assays of purinergic receptor function and this should be noted. Consider discussing changes in intracellular Ca²⁺ buffers that could occur with age and might alter intracellular Ca²⁺ responses.

Minor Comments

Line 42. State in Abstract that supporting cells were studied in the apical turn of the cochlea

Line 86. State "Glowatzki" et al. 2006

Lines 111-112. The observation that ATP is present in nanomolar concentrations in the cochlear fluids is mentioned several times throughout the paper (Munoz et al.). Include here that the purinergic receptors are found on the endolymphatic surface of cells and that Ca²⁺ signals have previously been shown to propagate from apex to base in supporting cells (Rabbitt and Holman 2021).

Line 420-421. Could provide more details here of the age-related changes seen in hair cells in the papers by Jeng et al.

Fig. 1C is labelled as 8 cells, but 9 data points are visible. ABR changes are evident at 20 months in 6N mice, but it would be helpful to state time course of hearing loss in the different mice strains in Methods.

Fig 4A seems to indicate large Ca²⁺ responses in the hair cells as well as supporting cells. If so, this should be mentioned.

Referee #2:

This paper examines the functional expression of three P2Y purinergic metabotropic receptors across development into adulthood with a strong emphasis on their possible role in age-related hearing loss (ARHL). The evidence that these receptors may be related to age-related changes mainly rests on data from other tissues, thus making this a valid research question for the cochlea and hearing. Three genetic mouse strains are used, two of which show marked ARHL (6N and 6N-Repaired), the other (C3H) shows good hearing up to at least 20 months, as shown in Fig. 1. Experiments involve immunofluorescence labelling, auditory brainstem responses and ratiometric calcium imaging using membrane-permeant Fura-2. The principal finding is that for the two mouse strains exhibiting ARHL average intracellular calcium responses get larger between 12 months and 20 months, including larger maxima. This follows an initial reduction in calcium responses between P7 and 1-2 months, which is apparent in all three mouse strains. The good-hearing mouse strain also shows a slight increase in average calcium response between 12 months and 20 months, but this was found to be non-significant. Additional experiments identify the purinergic receptors upregulated in the aged mice (P2Y2 and P2Y4). The Discussion focuses on the possible role of the upregulation of the P2Ys, loosely pointing towards a protective mechanism against cellular damage or promoting structural remodelling. The paper will be of interest to hearing physiologists or those focusing on purinergic signalling and functional outcomes.

While the main experimentally-derived points, as outlined above, are well made with good evidence, there some mismatches between data and conclusion (or predictions) that warrant attention.

1. The immunofluorescence data for 6N mice (Fig. 2) seems to me to be at odds with the authors interpretation that 'P2Y1, P2Y2 and P2Y4 were highly expressed in the supporting cells of the GER during pre-hearing stages of development (P7), but became largely down-regulated in supporting cells of young adult mice (1 month). However, in the aged cochlea, the puncta-like expression of P2Y2 and P2Y4, but not P2Y1, reappeared in the supporting cells'. Looking at the top view data (see below) there's brighter labelling at 1 month compared to P7 in all the P2Ys, and for P2Y1 1 month labelling looks brighter than aged:

P2Y1 at P7, 1 month, aged = weak, strong, medium labelling respectively

P2Y2 at P7, 1 month, aged = very weak, medium, strong

P2Y4 at P7, 1 month, aged = very weak, medium, strong

Looking at side view data P7 labelling does look brighter and more conspicuous than 1 month for all P2Ys in line with the authors conclusion, but labelling in the aged is weak to medium, so not obviously stronger than at P7.

For the good hearing C3H (Fig. 3), the authors report a similar pattern of labelling (to 6N) over developmental stages P7 to 1 month, but less strong labelling for aged mice for P2Y2 and P2Y4. I can see this pattern in the side view data, but for top view:

P2Y1 at P7, 1 month, aged = medium, medium, strong

P2Y2 at P7, 1 month, aged = very weak, weak, strong

P2Y4 at P7, 1 month, aged = weak, medium, strong

So the general pattern here looks like aged > 1 month {greater than or equal to} P7. It just needs a little more direction for the reader, and perhaps a tweak to the figures to allow the reader to see what I assume the authors are seeing. I can't really understand why the top and lateral views appear so different in terms of green labelling/brightness, and the arrows (that are not mentioned in text/legends) don't provide any help.

2. Discussion lines 524-528. The 'recapitulate' idea was described earlier in the paper (lines 421-423), but it is probably best suited as a take home message here, so not needed earlier. The authors ideas in these final lines may well turn out to be true, but I don't fully understand why they can discount the more negative conclusion e.g. that this P2Y upregulation might itself be part of ARHL. Given the state of current knowledge it would be best to highlight both possibilities at the end, then perhaps lean towards the positive as at present.

Minor points

1. Title: no need for 'the'.
2. Intro 70-75. These two sentences would actually work well as one e.g. 'So far.....innervations, however little is known....ageing.' The 8 citations appear to produce the break, but they could be placed at the end, or reduced in number?
3. Intro 76-80. This detail comes up again on lines 263-271. My suggestion would be to remove it from the Introduction.
4. Methods line 150. Provide a value for threshold in μV .
5. Methods lines 235-236. Some of the data look skewed with a high value tail, normal distribution?
6. Results 289-290. Give indication of mouse strain used.
7. Results line 299. Give indication of Munoz et al estimate of ATP in cochlear fluids here, at first mention (rather than later on).
8. Results line 311. Prehearing stages in Fig. 5 = A,E,I rather than the stated A-D.
9. Results line 341. In 'comparable' I think you mean 'reasonably close' although there is still a x5-10 difference.
10. Discussion line 437. Downregulation of P2Y is not a conclusion of the Huang et al (2010) paper although it might be possible to discern a pattern from the figures or table. The Sirko paper examines calcium waves in the adult cochlea, so again this downregulation is not a conclusion of the paper.
11. Discussion line 442. 'best mimics' = approaches.
12. Discussion line 462-464. What about being more reader-centric, rather than author-centric, with numbers of citations - which reference(s) would a reader need to consult to get a good idea for 2nd messenger diffusion within the cochlea?
13. Discussion line 479. Give an indication of when 'later in life' occurs.
14. Discussion lines 520-521. Missing detail - efferents to IHCs reform in aged animals.
15. Fig. 5 N-P. Its hard to discern the means and SDs, they seem lost among the data points.
16. Fig. 9C. How does the frequency compare to the earlier developmental ATP-triggered calcium waves in cochlear explants?

END OF COMMENTS

Confidential Review

04-May-2023

JP-RP-2023-284980

Age-related changes in P2Y receptor signalling in the mouse cochlear supporting cells

We thank the Reviewers for their numerous constructive and very helpful comments, which have helped us to strengthen the manuscript. Line numbers refers to: Hool et al 2023_Revised_Changes Highlighted.pdf

One of the major issues from both Reviewers was the very unclear immunostaining experiments in the submitted Figures 2 and 3. We realised that this issue was primarily caused by the compression of the pdf file produced by the online system, which almost completely removed the puncta-like labelling from the images. We apologise for not having spotted this during the submission. To avoid the same issue from happening, we have also included the improved immunolabelling Figures with the P2Y puncta quantification at the end of this document.

Following the excellent advice from the Reviewers, we have also largely improved the readability of the images by adding the brightfield images with annotations that highlight the key features of the cochlear epithelia. Because of the addition of several new panels, we have re-organised the imaging data into three separate Figures (Figures 2: P2Y₁; Figures 3: P2Y₂; Figures 4: P2Y₄).

Reviewing Editor:

This manuscript provides several new observations on the inner mechanisms that result in age-related hearing loss. Both reviewers found the data interesting, insightful and will be of broad interest to the hearing sciences research community. Both reviewers found much of the data very supportive of the conclusions. However, both reviewers have concerns with confocal imaging experiments as these data appear to be weakly supportive of conclusions drawn. To strengthen conclusions in the manuscript, the authors need to address the confocal imaging data concerns. Finally, the authors should carefully revise and rewrite their manuscript based on the positive criticism provided by the reviewers.

We believe that all criticisms raised by both Reviewers have been addressed in the revised manuscript.

Ethics: Please add statement on unlimited access to food and water.

Done (ln. 118).

Referee #1:

This manuscript by Hool et al. aims to describe the localization and functional properties of metabotropic purinergic P2Y receptors in supporting cells of the apical mouse cochlea with age. Cochlea tissue from mouse models showing early (6N) or late onset (C3H and 6N repaired) hearing loss were immunolabeled for purinergic receptors and compared. Ratiometric calcium imaging experiments in explants show supporting cells respond to extracellular application of ATP and purinergic receptor subtypes mediating Ca²⁺ responses are characterized with various agonists and antagonists. The authors find that expression of supporting cell P2Y₁, P2Y₂ and P2Y₄ receptors decreases during postnatal cochlear maturation as described previously (consistent with the disappearance of Kolliker's organ). New findings show apparent upregulation of P2Y₂ and P2Y₄-mediated responses at older ages. The authors speculate that since the upregulation is greater in early onset hearing loss mice receptor changes may be protective to hair cells, although the mechanisms are not clear. Overall, the calcium imaging experiments with pharmacological characterization of purinergic responses in support cells of aging mice provide interesting new data. The conclusions from immunohistochemistry data are less well supported.

1. The cartoons showing distributions of cell types in immature and mature cochlea in Fig. 2 are helpful for orientation, but visualizing cell layers and arrangements in the confocal images is problematic. The authors should consider using cellular markers to label hair cells (anti-myosin 7A) and supporting cells (anti-Sox2). Immunolocalization of purinergic receptors in supporting cells of the medial compartment is unclear (Figs 2 and 3). Z-stack images (number of stacks and image acquisition details are not given) at different ages are shown and changes in receptor expression with age are claimed. But in some cases, receptor labeling looks the same as the actin labeling of hair bundles (e.g. P2Y2 signal and phalloidin signal in Fig. 3B). Also, phalloidin labelled hair bundles in Fig 2B middle image P7 are only partly visible. Since only representative images (with no quantification of immunofluorescence) are shown at the different ages, conclusions on age-related changes seem overstated.

As mentioned in the above summary, the compression of the pdf file produced by the JPhysiol online system almost completely removed the puncta-like labelling from the images. This would have made very difficult to distinguish any P2Y puncta. To further increase the contrast, we now show the P2Y puncta in white (see revised Figures 2-4 that are also appended at the end of this document).

We considered the possible use of Sox2 and Myo7a antibodies, but unfortunately those that work in our hands are all raised in rabbit, which is the same of the P2YRs. However, we have proposed an alternative solution to improve the clarity of the cellular structure by adding the brightfield images of the cochlea at different ages. The advantage of this is that the borders of the different cell types within the GER and IS can be easily identified, highlighted, and annotated. These new images are now included in the revised Figure 2. We did not replicate the brightfield images in the New Figure 3 and 4 to avoid unnecessary duplications.

We have now included more details on the z-stack acquisition (ln. 168).

We have also attempted to quantify the number of P2Y receptor puncta from the z-stack images using ZEISS Arivis Scientific Image Analysis software. However, we could not find a common intensity threshold value that could be applied to all conditions (between different ages, mouse strains and P2Y receptors). This made it impossible to perform a reliable and reproducible quantification of the puncta, which is the reason we decided to keep the immunostaining data as a qualitative assessment. We have clearly stated this in the revised manuscript (ln. 288-299). Nevertheless, the combination of Ca^{2+} imaging with the specific P2Y pharmacology (Figures 7-11) provides a robust and reliable functional readout of the immunostaining experiments.

2. As shown in Fig. 2A and described in Results (lines 263-271) there are several different types of supporting cell in the medial compartment region. Were attempts made to classify support cells by type when examining Ca^{2+} responses? In this regard it could be helpful if a brightfield image (mentioned on line 217) of the preparation could be shown along with the manually placed ROIs in Fig. 4A.

We would like to thank the reviewer for this suggestion, which we have tried to address. We now include an additional panel A in the revised Figure 5 (previously submitted Figure 4) showing the raw fura-2 fluorescence image, alongside the false-colour images highlighting Ca^{2+} changes), where the individual IHCs and supporting cells in the GER can easily be identified. Although brightfield images obtained with our spinning disk confocal microscope are useful for ROI placement when they can be zoomed in and out

Fig. A. Brightfield image of the immature GER.

on a screen alongside fluorescence images, it is difficult to distinguish the individual supporting cells from the focal plane we do imaging from when the images are viewed in a static figure (especially on paper, see Fig. A).

Therefore, we have sought to address the reviewer's concerns as follows:

1) We now include an additional panel A in the revised Figure 5 showing the raw fura-2 fluorescence image, where the individual IHCs and supporting cells in the GER can easily be identified. We believe that this solution is better for visualisation purposes.

2) We have also realised that the sample in the submitted version was from a P8 cochlea, which does not match the typical experiments used in the submitted Figure 5 (P7: now Figure 6). Therefore, we have now included an example at P7 (see new Figure 5).

3) Although we have not classified the supporting cells by type when looking at Ca²⁺ responses, in this work we have limited our analysis to the SCs close to the IHCs both in development and in mature stages. We have now provided a better identification of the investigated cell types by providing a direct link between the immunolabelling (new Figures 2-4) and the Ca²⁺ imaging experiments. In the revised Figure 6 (previously submitted Figure 5) we have included a new panel A that includes the same brightfield images used for immunostaining (Figure 2), but with the specific region of the GER and IS used for the Ca²⁺ imaging experiments highlighted by dashed white lines.

3. How was the "average Ca²⁺ response" measured (Fig. 5N)? Please describe the averaging technique in Methods. It's interesting that the maximum Ca²⁺ response was highly variable in the 6N mice at P7-8 (Fig. 5O), could you speculate why this might be? Also, a description of how "oscillation frequency" was measured for Fig. 5P would be helpful and describe how oscillations differ from repetitive Ca²⁺ spikes (lines 331-332).

We have added more information about quantification of the Ca²⁺ signal (max, average and oscillations) in the methods (ln. 229-236). We have also better defined the meaning of "calcium spikes" (ln. 314-317).

The variability of Ca²⁺ responses at P7-8 could be due to differences in number of purinergic receptor subtypes in different cells or differences in sensitization of purinergic receptors due to ongoing spontaneous Ca²⁺ activity in immature cochleae. Nevertheless, the responses were not different between the 6N and C3H mice (ln. 333-337).

4. The changes in P2Y signals with age are interesting, but their role in age-related hearing loss is unresolved. The Discussion of age-related changes (p17-18) is rather long and inconclusive. Suggest substantially shortening this section by deleting description of age-related changes in purinergic signaling in other systems. Ca²⁺ responses shown here are indirect assays of purinergic receptor function and this should be noted. Consider discussing changes in intracellular Ca²⁺ buffers that could occur with age and might alter intracellular Ca²⁺ responses.

We agree with this Reviewer that the specific reason for the changes in P2Y signalling need further investigation, which we have highlighted in the Discussion. We would also like to point out that even in the developing cochlea, which has been studied for many years, the role of these P2Y receptors is also not fully understood. Our study provides the first indication that the physiological responses of the supporting cells are likely to change in the aged cochlea, and that P2Y receptors are involved.

To address the Reviewer's points, we have removed the reference to the changes in purinergic signalling in other systems.

We have also expanded the section highlighting the very likely possibility that other mechanisms change with age (ln. 481-487).

Minor Comments

Line 42. State in Abstract that supporting cells were studied in the apical turn of the cochlea

Done (ln. 41-43)

Line 86. State "Glowatzki" et al. 2006

Done

Lines 111-112. The observation that ATP is present in nanomolar concentrations in the cochlear fluids is mentioned several times throughout the paper (Munoz et al.). Include here that the purinergic receptors are found on the endolymphatic surface of cells and that Ca²⁺ signals have previously been shown to propagate from apex to base in supporting cells (Rabbitt and Holman 2021).

We agree with the reviewer that the articles from Munoz et al. were cited so many times throughout the manuscript, which is very repetitive. We have removed several of these repetitive statements.

Thank you for excellent suggestion about better explain the direction in which the purinergic signalling travels in the supporting cells, which we now include in the Introduction (ln. 86-91).

Line 420-421. Could provide more details here of the age-related changes seen in hair cells in the papers by Jeng et al.

As requested, we have included a more detailed explanation in the revised Discussion. Because the initial paragraph is mainly used to provide an overall take on message of the paper, the additional text is included toward the end of the Discussion (ln. 541-544).

Fig. 1C is labelled as 8 cells, but 9 data points are visible. ABR changes are evident at 20 months in 6N mice, but it would be helpful to state time course of hearing loss in the different mice strains in Methods.

Thank you for spotting this mistake. We have inverted the "n" number for the aged mice between 6N (n = 9) and C3H (n = 8). We have now corrected this mistake in the revised Figure 1. As requested, we have also added more details on the progression of hearing of the different strains in the Methods (ln. 126-134).

Fig 4A seems to indicate large Ca²⁺ responses in the hair cells as well as supporting cells. If so, this should be mentioned.

Because the IHCs are not a direct topic of this paper, we have added a statement in the Figure legend, and referred to the literature (ln. 860-862).

Referee #2:

This paper examines the functional expression of three P2Y purinergic metabotropic receptors across development into adulthood with a strong emphasis on their possible role in age-related hearing loss (ARHL). The evidence that these receptors may be related to age-related changes mainly rests on data from other tissues, thus making this a valid research question for the cochlea and hearing. Three genetic mouse strains are used, two of which show marked ARHL (6N and 6N-Repaired), the other (C3H) shows good hearing up to at least 20 months, as shown in Fig. 1. Experiments involve immunofluorescence labelling, auditory brainstem responses and ratiometric calcium imaging using membrane-permeant Fura-2. The principal finding is that for the two mouse strains exhibiting ARHL average intracellular calcium responses get larger between 12 months and 20 months, including larger maxima. This follows an initial reduction in calcium responses between

P7 and 1-2 months, which is apparent in all three mouse strains. The good-hearing mouse strain also shows a slight increase in average calcium response between 12 months and 20 months, but this was found to be non-significant. Additional experiments identify the purinergic receptors upregulated in the aged mice (P2Y2 and P2Y4). The Discussion focuses on the possible role of the upregulation of the P2Ys, loosely pointing towards a protective mechanism against cellular damage or promoting structural remodelling. The paper will be of interest to hearing physiologists or those focusing on purinergic signalling and functional outcomes.

While the main experimentally-derived points, as outlined above, are well made with good evidence, there are some mismatches between data and conclusion (or predictions) that warrant attention.

1. The immunofluorescence data for 6N mice (Fig. 2) seems to me to be at odds with the authors' interpretation that 'P2Y1, P2Y2 and P2Y4 were highly expressed in the supporting cells of the GER during pre-hearing stages of development (P7), but became largely down-regulated in supporting cells of young adult mice (1 month). However, in the aged cochlea, the puncta-like expression of P2Y2 and P2Y4, but not P2Y1, reappeared in the supporting cells'. Looking at the top view data (see below) there's brighter labelling at 1 month compared to P7 in all the P2Ys, and for P2Y1 1 month labelling looks brighter than aged:

P2Y1 at P7, 1 month, aged = weak, strong, medium labelling respectively

P2Y2 at P7, 1 month, aged = very weak, medium, strong

P2Y4 at P7, 1 month, aged = very weak, medium, strong

Looking at side view data P7 labelling does look brighter and more conspicuous than 1 month for all P2Ys in line with the authors' conclusion, but labelling in the aged is weak to medium, so not obviously stronger than at P7.

For the good hearing C3H (Fig. 3), the authors report a similar pattern of labelling (to 6N) over developmental stages P7 to 1 month, but less strong labelling for aged mice for P2Y2 and P2Y4. I can see this pattern in the side view data, but for top view:

P2Y1 at P7, 1 month, aged = medium, medium, strong

P2Y2 at P7, 1 month, aged = very weak, weak, strong

P2Y4 at P7, 1 month, aged = weak, medium, strong

So the general pattern here looks like aged > 1 month {greater than or equal to} P7. It just needs a little more direction for the reader, and perhaps a tweak to the figures to allow the reader to see what I assume the authors are seeing. I can't really understand why the top and lateral views appear so different in terms of green labelling/brightness, and the arrows (that are not mentioned in text/legends) don't provide any help.

As mentioned in the first general paragraph above, we have completely missed the poor rendering of the immunostaining figures present in the pdf generated by the JPhysiol online system. We apologise for this oversight. To avoid the same issue, we have now included the improved immunolabelling figures at the end of this document.

Nevertheless, the comments from this Reviewer have instigated us to provide a better visual understanding of the immunostaining data. To better highlight the key features of the cochlear epithelia, we have added the brightfield images at different ages with annotations of the relevant cellular structures. The advantage of this is that the borders of the different cell types within the GER and IS can be easily identified, highlighted, and annotated. It should be now clear that we are

investigating the puncta-like labelling in the supporting cells of the GER and IS, and not that of the pillar cells that were not investigated in the Ca^{2+} imaging experiments (orange arrows in the revised Figure 2). These new images are now included in the revised Figure 2. We did not replicate the brightfield images in the new Figure 3 and 4 to avoid unnecessary duplications.

As mentioned to Reviewer 1, we have also attempted to quantify the number of P2Y receptor puncta from the z-stack images using ZEISS Arivis Scientific Image Analysis software. However, we could not find a common intensity threshold value that could be applied to all conditions (between different ages, mouse strains and P2Y receptors). This made it impossible to perform a reliable and reproducible quantification of the puncta, which is the reason we decided to keep the immunostaining data as a qualitative assessment. We have clearly stated this in the revised manuscript (ln. 288-299). Nevertheless, the combination of Ca^{2+} imaging with the specific P2Y pharmacology (Figures 7-11) provides a robust and reliable functional readout of the immunostaining experiments.

2. Discussion lines 524-528. The 'recapitulate' idea was described earlier in the paper (lines 421-423), but it is probably best suited as a take home message here, so not needed earlier. The authors ideas in these final lines may well turn out to be true, but I don't fully understand why they can discount the more negative conclusion e.g. that this P2Y upregulation might itself be part of ARHL. Given the state of current knowledge it would be best to highlight both possibilities at the end, then perhaps lean towards the positive as at present.

This is a very reasonable hypothesis/request, which we have now implemented both in the initial paragraph (ln. 438-441), the aim of which is to provide a self-contained overview of the entire study, and also at the end of the Discussion (ln. 544-549).

Minor points

1. Title: no need for 'the'.

Done. Thank you.

2. Intro 70-75. These two sentences would actually work well as one e.g. 'So far.....innervations, however little is known....ageing.' The 8 citations appear to produce the break, but they could be placed at the end, or reduced in number?

Done

3. Intro 76-80. This detail comes up again on lines 263-271. My suggestion would be to remove it from the Introduction.

Thank you. Sentence removed from the Introduction.

4. Methods line 150. Provide a value for threshold in μV .

The sound thresholds are measured in decibels. We have now included this in the Methods (ln. 148-150).

5. Methods lines 235-236. Some of the data look skewed with a high value tail, normal distribution?

We thank the reviewer for this observation. Calcium responses measurements are indeed slightly skewed with a high value tail. We therefore re-computed the statistical significance using non-parametric tests for quantification of Ca^{2+} signals. We amended the Methods section (ln. 233-240) and the statistical analysis throughout the Result section and figure legends.

6. Results 289-290. Give indication of mouse strain used.

This is a general statement that applies to all mouse strains, which we now clearly state (ln. 302-304). However, we have also included the mouse strains in the legend of Figure 5.

7. Results line 299. Give indication of Munoz et al estimate of ATP in cochlear fluids here, at first mention (rather than later on).

In response to Reviewer 1, we have now removed this sentence since it was duplicated several times throughout the manuscript.

8. Results line 311. Prehearing stages in Fig. 5 = A,E,I rather than the stated A-D.

The Figures has been revised, so the labels are now different, but we have corrected the mistake.

9. Results line 341. In 'comparable' I think you mean 'reasonably close' although there is still a x5-10 difference.

Changed (ln. 357-361).

10. Discussion line 437. Downregulation of P2Y is not a conclusion of the Huang et al (2010) paper although it might be possible to discern a pattern from the figures or table. The Sirko paper examines calcium waves in the adult cochlea, so again this downregulation is not a conclusion of the paper.

Besides the table and figures, the Huang et al (2010) paper reports that “*Compared with the neonatal tissue, the other supporting cells showed down-regulation of P2Y receptor expression. P2Y4 expression was maintained in the inner sulcus region (Fig. 6c), albeit at a lower level (compare with Fig. 3c), and also developed in the outer sulcus cell region (Fig. 6c)*”. Therefore, we believe that the citation is justified in our Discussion.

Regarding the Sirko et al. (2019) paper, they report that “*Consistent with this conclusion [a change in cell properties in the inner sulcus], there was no slow second ATP-activated current component in the adult IS cells characteristic of P2Y receptor activation. ATP signalling in the IS could thus play a different physiological role in the adult and may be reflected in the different Ca²⁺ wave properties*”. Even though this conclusion points to a change in P2Y receptors after the onset of hearing, we agree with the reviewer that it does not necessarily imply a downregulation of P2Y receptors in the adult cochlea.

To better reflects the points listed above, we have now revised this section of the Discussion (ln 481-490).

11. Discussion line 442. 'best mimics' = approaches.

Changed (ln. 481-484).

12. Discussion line 462-464. What about being more reader-centric, rather than author-centric, with numbers of citations - which reference(s) would a reader need to consult to get a good idea for 2nd messenger diffusion within the cochlea?

Thank you, useful thought. Sometimes we forget that by trying to give credit to the large body of work we make the job of the reader more challenging. We have now only cited the key articles (ln. 487-590).

13. Discussion line 479. Give an indication of when 'later in life' occurs.

We have now included the age range (ln. 500-503)

14. Discussion lines 520-521. Missing detail - efferents to IHCs reform in aged animals.

Thank you. We have now included the link with the IHCs (ln. 538-540).

15. Fig. 5 N-P. Its hard to discern the means and SDs, they seem lost among the data points.

We have revised the Figure to make the mean and SD more visible.

16. Fig. 9C. How does the frequency compare to the earlier developmental ATP-triggered calcium waves in cochlear explants?

Considering that comparing submitted Figure 9 (now Figure 10) is investigating the effect of UTP, we are assuming that this Reviewer is asking to compare the Ca^{2+} signals in the supporting cells between pre-hearing and aged mice. We are not sure about the rationale of comparing UTP to ATP responses. Therefore, we have calculated the frequency of UTP induced Ca^{2+} oscillations and reported it in the text (ln. 937-942).

Figure 2

A Side view of the Immature Organ of Corti

Side view of the Mature Organ of Corti

B 6N Top view

D 6N Top view

F 6N Side view

C C3H Top view

E C3H Top view

G C3H Side view

Figure 3

Figure 4

Dear Professor Marcotti,

Re: JP-RP-2023-284980R1 "Age-related changes in P2Y receptor signalling in mouse cochlear supporting cells" by Sarah A Hool, Jing-Yi Jeng, Dan Jagger, Walter Marcotti, and Federico Ceriani

Thank you for submitting your manuscript to The Journal of Physiology. It has been assessed by a Reviewing Editor and by 2 expert referees and we are pleased to tell you that it is acceptable for publication following satisfactory revision.

REVISION CHECKLIST:

We look forward to receiving your revised submission.

Yours sincerely,

Katalin Toth
Senior Editor
The Journal of Physiology

EDITOR COMMENTS

Reviewing Editor:

The authors have done an excellent job of responding to the previous critiques. Please edit the manuscript accordingly in response to the comments of Reviewer#1 and Reviewer#2.

REFEREE COMMENTS

Referee #1:

The authors have responded adequately to the comments in the previous review. The immunofluorescence images of cochlea are now presented at better resolution and corresponding DIC images provided. I have a couple of remaining queries and comments:

1. Line 137 ABR Methods. Please clarify if mice were euthanised following ABR measurements or if they were allowed to recover following anesthesia.
2. Line 457 delete "the" before rat.
3. Line 524 should read "act" not "acts"

Referee #2:

The impact, insight, originality, design and validity are as for the previous version, or very close to.

The authors have modified the paper as suggested. I can now follow the confocal images in line with the authors narrative, although clearly the punctate P2Y labelling is very fine when present, particularly in top view (Fig. 2-4), and (as the authors state) can get obscured when converted to pdf format. They will need to keep an eye on this. There are a few instances where singular/plural alterations are needed, and a few other minor changes. Inevitably, on second reading a few addition questions crop up, but nothing that should present a major hurdle.

In the following the line numbers (in brackets) relate to the highlighted version of the paper.

1. (70-74). These two sentences would actually work well as one e.g. 'So far.....innervations, however little is known.....ageing.' It would be better as one sentence since the refs, how moved as suggested, still relate to the two sentences as currently written.
2. (86-90). Sentence repeated.
3. (104). Better as 'displaying varying degrees'
4. (150). Provide a value for threshold in uV. I was referring to the 'was visible', so what's the minimum visible response amplitude in uV used to estimate threshold?
5. (200). Better as 'Fura-2 fluorescence was excited at two alternating wavelengths.'
6. (222). The background subtraction was done by subtracting an average background value (no Fura-2) from each pixel, or by subtracting actual background values on a pixel by pixel basis?
7. (225). Define the various Rs and avoid a mix of subscript/no subscript.
8. (233-4). The detail in brackets is unnecessary.
9. (236) 'inspected manually to remove any spurious peak' and reanalysed / recalculated? (Also peaks better than peak here.)
10. (239) The Wilcoxon signed-rank test is for related paired data. The Mann-Whitney U test is appropriate for unrelated non-parametric data, e.g. where there are 2 columns of data, but in no particular order or no strict pairing.
11. (249) replace 'laboratory' with a reference.
12. (260) 'only slightly so' would be better as 'less so', although the p values probably provide enough information on their own.
13. (292) typo Fig. 2H.
14. (326) do you need to refer to Fig.6 B,E? Also, regarding Fig. 6, can the authors comment on the clear and interesting alignment of peaks mid-application as in B,F,L? How does the sequence of records in these panels reflect the position of the ROIs in the image?
15. (394). 'no age-related relationship': I agree nothing here equivalent to Fig.6, but Fig.9A-C show some evidence of age affecting the responses and you don't compare P7-8 & 1-2 months in the average and maximum response plots.
16. (435). In place of 'Considering' put 'Note that', this allows the sentence to stand on its own.
17. (524 and 538) Change 'acts' to act and 'recapitulate' to recapitulates.
18. (544). The age-related changes in the efferent system (e.g. reforming synapses onto IHCs) are unlikely to be part of a progressive deterioration of the mammalian cochlea with age but they could be a response to it, however with up-regulation of P2Y receptors, I'm not sure the force of the argument is the same. But you could explain the case for the age-related changes in efferent contacts, and then suggest a similar role for up-regulation P2Ys.
19. (901). Give a value for n
20. (917). What does 'left' refer to?
21. (920). E is maximum, F is average (see Fig. 9).

END OF COMMENTS

Referee #1:

The authors have responded adequately to the comments in the previous review. The immunofluorescence images of cochlea are now presented at better resolution and corresponding DIC images provided. I have a couple of remaining queries and comments:

1. Line 137 ABR Methods. Please clarify if mice were euthanised following ABR measurements or if they were allowed to recover following anaesthesia.

Thank you for spotting this important omission. We have now included a statement about the anaesthesia in the "Ethical Statement" (ln. 118-124).

2. Line 457 delete "the" before rat.

Done.

3. Line 524 should read "act" not "acts"

Done.

Referee #2:

The authors have modified the paper as suggested. I can now follow the confocal images in line with the authors narrative, although clearly the punctate P2Y labelling is very fine when present, particularly in top view (Fig. 2-4), and (as the authors state) can get obscured when converted to pdf format. They will need to keep an eye on this. There are a few instances where singular/plural alterations are needed, and a few other minor changes. Inevitably, on second reading a few addition questions crop up, but nothing that should present a major hurdle.

In the following the line numbers (in brackets) relate to the highlighted version of the paper.

1. (70-74). These two sentences would actually work well as one e.g. 'So far.....innervations, however little is known....ageing.' It would be better as one sentence since the refs, how moved as suggested, still relate to the two sentences as currently written.

We have now merged the two sentences (ln. 70-74).

2. (86-90). Sentence repeated.

Thank you for spotting this error. The first incomplete sentence has been deleted.

3. (104). Better as 'displaying varying degrees'.

Done.

4. (150). Provide a value for threshold in uV. I was referring to the 'was visible', so what's the minimum visible response amplitude in uV used to estimate threshold?.

As commonly done in the field, auditory thresholds are defined as the lowest sound pressure level measured in decibel (dB) (ln. 153-155).

5. (200). Better as 'Fura-2 fluorescence was excited at two alternating wavelengths..'

Done.

6. (222). The background subtraction was done by subtracting an average background value (no Fura-2) from each pixel, or by subtracting actual background values on a pixel by pixel basis?

We subtracted an average background value, not a pixel-by-pixel image. We have clarified this in the Methods. (ln. 227-229).

7. (225). Define the various Rs and avoid a mix of subscript/no subscript.

Done (ln. 230-234).

8. (233-4). The detail in brackets is unnecessary.

We prefer to keep this description because it refers to the exact algorithm we used for the analysis.

9. (236) 'inspected manually to remove any spurious peak' and reanalysed / recalculated? (Also peaks better than peak here.)

Corrected.

10. (239) The Wilcoxon signed-rank test is for related paired data. The Mann-Whitney U test is appropriate for unrelated non-parametric data, e.g. where there are 2 columns of data, but in no particular order or no strict pairing.

This is correct. We used the Wilcoxon signed-rank test for related paired data (Fig. 8B, Fig.9H) and the Wilcoxon rank-sum test, also known as Mann-Whitney U test or Wilcoxon-Mann-Whitney test for unrelated non-parametric data (e.g., Fig.6O-P). See also the “Statistical analysis” section of the Methods (ln 245-251).

11. (249) replace 'laboratory' with a reference.

Done (ln. 255-257).

12. (260) 'only slightly so' would be better as 'less so', although the p values probably provide enough information on their own.

Done.

13. (292) typo Fig. 2H.

Correct. We have removed the reference to Fig. 2H.

14. (326) do you need to refer to Fig.6 B,E? Also, regarding Fig. 6, can the authors comment on the clear and interesting alignment of peaks mid-application as in B,F,L? How does the sequence of records in these panels reflect the position of the ROIs in the image?

We have removed the reference to Fig. 6B,E. Regarding the synchronised calcium signals, these are calcium waves propagating through several cells, and thus appearing as an (almost) simultaneous peak in several calcium traces. These events occur spontaneously in the developing cochlea, but can also be elicited by exogenous ATP application (Anselmi et al. 2008). We observed these events also in post-hearing cochleae, (e.g., the example in Fig. 6L), although they appear to be less common than in the immature cochlea. We added a comment on this point in Fig.6 legend (ln. 879-881).

The ROIs are usually placed sequentially in the image, so subsequent traces in Fig.6 belong to spatially adjacent ROIs.

15. (394). 'no age-related relationship': I agree nothing here equivalent to Fig.6, but Fig.9A-C show some evidence of age affecting the responses and you don't compare P7-8 & 1-2 months in the average and maximum response plots.

We added “after the onset of hearing” (ln. 400-402) to clarify what we mean by “no age-dependent relationship. We focused the comparison between adult and aged mice because it the only age range we used to discuss the key differences with the other purinergic receptors.

16. (435). In place of 'Considering' put 'Note that', this allows the sentence to stand on its own.

Done.

17. (524 and 538) Change 'acts' to act and 'recapitulate' to recapitulates.

Done.

18. (544). The age-related changes in the efferent system (e.g. reforming synapses onto IHCs) are unlikely to be part of a progressive deterioration of the mammalian cochlea with age but they could be a response to it, however with up-regulation of P2Y receptors, I'm not sure the force of the argument is the same. But you could explain the case for the age-related changes in efferent contacts, and then suggest a similar role for up-regulation P2Ys.

We have changed the sentence to highlight the concept of “consequence” of the progressive deterioration (ln. 549-552).

19. (901). Give a value for n

Done (ln. 906-908).

20. (917). What does 'left' refer to?

The word has been removed.

21. (920). E is maximum, F is average (see Fig. 9).

Corrected.

Dear Dr Marcotti,

Re: JP-RP-2023-284980R2 "Age-related changes in P2Y receptor signalling in mouse cochlear supporting cells" by Sarah A Hool, Jing-Yi Jeng, Dan Jagger, Walter Marcotti, and Federico Ceriani

We are pleased to tell you that your paper has been accepted for publication in The Journal of Physiology.

IMPORTANT

We seem to be missing a figure legend to accompany your abstract figure. Please can you send an abstract figure legend to Diana at the Editorial Office as soon as possible, please: jp@physoc.org
Many thanks!

Authors should note that it is too late at this point to offer corrections prior to proofing. The accepted version will be published online, ahead of the copy edited and typeset version being made available. Major corrections at proof stage, such as changes to figures, will be referred to the Editors for approval before they can be incorporated. Only minor changes, such as to style and consistency, should be made at proof stage. Changes that need to be made after proof stage will usually require a formal correction notice.

Yours sincerely,

Katalin Toth
Senior Editor
The Journal of Physiology

P.S. - You can help your research get the attention it deserves! Check out Wiley's free Promotion Guide for best-practice recommendations for promoting your work at www.wileyauthors.com/eeo/guide. You can learn more about Wiley Editing Services which offers professional video, design, and writing services to create shareable video abstracts, infographics, conference posters, lay summaries, and research news stories for your research at www.wileyauthors.com/eeo/promotion.

IMPORTANT NOTICE ABOUT OPEN ACCESS: To assist authors whose funding agencies mandate public access to published research findings sooner than 12 months after publication, The Journal of Physiology allows authors to pay an Open Access (OA) fee to have their papers made freely available immediately on publication.

You can check if your funder or institution has a Wiley Open Access Account here: <https://authorservices.wiley.com/author-resources/Journal-Authors/licensing-and-open-access/open-access/author-compliance-tool.html>.

EDITOR COMMENTS

Reviewing Editor:

The authors have addressed all final comments.